# The Common Pile v0.1: An 8TB Dataset of Public Domain and Openly Licensed Text

Nikhil Kandpal[*1,2]   Brian Lester[*1,2]   Colin Raffel[*1,2,3]
Sebastian Majstorovic[4]   Stella Biderman[4]   Baber Abbasi[4]   Luca Soldaini[5]   Enrico Shippole[6]
A. Feder Cooper[†7]   Aviya Skowron[4]   John Kirchenbauer[8]   Shayne Longpre[9]   Lintang
Sutawika[4,10]   Alon Albalak[‡11]   Zhenlin Xu[12]   Guilherme Penedo[3]   Loubna Ben Allal[3]   Elie
Bakouch[3]   John David Pressman[4]   Honglu Fan[4,13]   Dashiell Stander[4]   Guangyu Song[4]   Aaron
Gokaslan[7]   Tom Goldstein[8]   Brian R. Bartoldson[14]   Bhavya Kailkhura[14]   Tyler Murray[5]

[1]University of Toronto   [2]Vector Institute   [3]Hugging Face   [4]EleutherAI   [5]The Allen Institute for
Artificial Intelligence   [6]Teraflop AI   [7]Stanford University   [8]University of Maryland, College Park
[9]MIT   [10]CMU   [11]Lila Sciences   [12]Independent   [13]poolside   [14]Lawrence Livermore National
Laboratory

## Abstract

Large language models (LLMs) are typically trained on enormous quantities of
unlicensed text, a practice that has led to scrutiny due to possible copyright in-
fringement and ethical concerns. Training LLMs on openly licensed text presents
a first step towards addressing these issues, but prior data collection efforts have
yielded datasets too small or low-quality to produce performant LLMs. To address
this gap, we collect, curate, and release the Common Pile v0.1, an eight terabyte
collection of openly licensed text designed for LLM pretraining. The Common Pile
comprises content from 30 sources that span diverse domains including research
papers, code, books, encyclopedias, educational materials, audio transcripts, and
more. Crucially, we validate our efforts by training two 7 billion parameter LLMs
on text from the Common Pile: Comma v0.1-1T and Comma v0.1-2T, trained on
1 and 2 trillion tokens respectively. Both models attain competitive performance
to LLMs trained on unlicensed text with similar computational budgets, such as
Llama 1 and 2 7B. In addition to releasing the Common Pile v0.1 itself, we also
release the code used in its creation as well as the training mixture and checkpoints
for the Comma v0.1 models.

## 1   Introduction

A critical stage of large language model (LLM) development is pretraining [73, 137, 143], where an
LLM is trained to predict the next token (i.e., word or subword unit) in a corpus of unstructured text.
Pretraining is widely regarded as the foundation for strong downstream performance, as it enables
LLMs to learn the structure of natural language [32, 111, 155] and accumulate a broad base of world
knowledge [134, 153]. In an effort to push the capabilities of LLMs, pre-training datasets have grown
steadily over time [144], with modern datasets containing trillions of tokens [133, 168, 194]. To meet
this increasing demand for pre-training data, the *de facto* approach has been to leverage the public
Internet as a source of text [58, 96, 109, 133, 143].

While the web provides a diverse and continuously growing supply of text, under most legal frame-
works, much of this content is protected by copyright. Yet, this text is routinely used to pretrain

---

[*]Equal contribution. For a list of author contributions, see Appendix A. [†]Work done while a graduate
student at Cornell University. [‡]Work done while at SynthLabs.

39th Conference on Neural Information Processing Systems (NeurIPS 2025) Track on Datasets and Benchmarks.

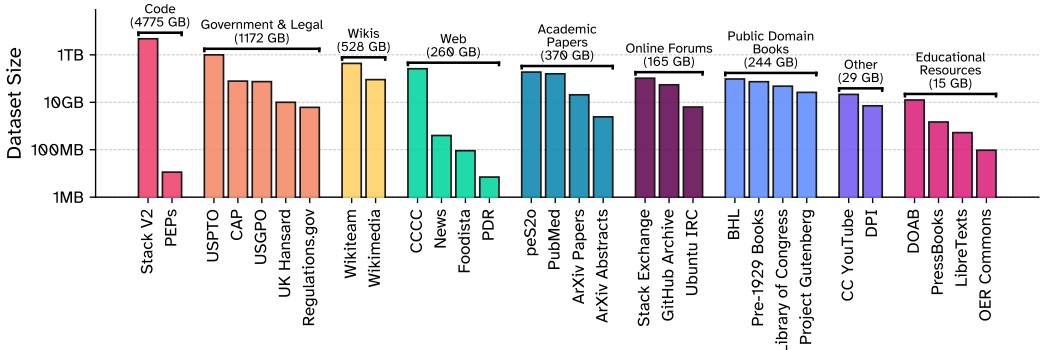

Figure 1: **The Common Pile is an 8TB dataset of openly licensed text curated from 30 diverse sources.** The sources comprising the Common Pile are shown above, categorized by textual domain.

LLMs, often without compensation to the creators of this content. Recent estimates suggest that compensating the authors of pre-training data, even at conservatively low wage rates, would cost billions of US dollars [83]. While copyright exemptions for text and data mining exist in some jurisdictions [70, 80, 93, 131, 157], many rights holders have objected to the uncompensated use of their work, resulting in numerous lawsuits against LLM developers [24, 192] that could carry financial damages in the billions [40, 97, 160]. Beyond questions of intellectual property (IP) law, the use of web-scraped data also raises ethical concerns [9], as content creators rarely explicitly consent to the downstream use of their work for LLM training. In fact, recent evidence suggests that many content owners may *not* consent to its use as LLM training data, as shown by a sharp mid-2023 increase in websites blocking AI crawlers [108], following growing awareness of web data being used to train models. Finally, while open models trained on publicly released pre-training datasets [18, 66, 104] support research into the study of learning dynamics [51, 77, 85], memorization [17, 22], data auditing [48, 129, 146], and more, the use of unlicensed training data heavily limits the ability of model trainers to share their datasets, and has previously resulted in DMCA takedowns of datasets such as the Pile [58].

The current landscape reflects a growing divide between LLM developers and content creators. We submit that a natural first step toward resolving this tension is to ask: *Is it possible to train performant language models using only public domain and openly licensed text?* We define "openly licensed" text as content that follows the Open Knowledge Foundation's Open Definition 2.1 (further detailed in section 2 and Appendix C), which refers to content where the copyright holder has granted explicit permission for the content to be freely accessed, used, modified, and shared for any purpose. Our primary contribution in this paper is to demonstrate that this is indeed possible by collecting, curating, and releasing *the Common Pile v0.1*, an 8TB dataset that—to our knowledge—constitutes the largest collection of openly licensed text to date. The Common Pile comprises 30 text sources (detailed in section 3), covering diverse domains including research publications, open-source code, government documents, historical books, educational resources, audio transcripts, and more. Crucially, we demonstrate that after appropriate filtering, deduplication, and reweighting, the Common Pile v0.1 can be used as the foundation for competitive LLMs. Specifically, we train Comma v0.1-1T and Comma v0.1-2T, a pair of 7-billion-parameter models with comparable performance to budget-matched models trained on unlicensed datasets such as Llama 1 and 2 7B. In the spirit of openness and transparency, we release the Common Pile v0.1, both Comma v0.1 models and their filtered and deduplicated pre-training dataset, and all data collection and processing code.

## 2 What do we mean by "openly licensed"?

Copyright law grants content creators certain rights, such as exclusive rights (with certain exceptions) to reproduce, distribute, and create derivatives of their original works. Although copyright laws vary across jurisdictions, original, (modestly) creative works (that are "fixed" in a tangible medium, such as physically or digitally [see, e.g., 1]) typically fall within the scope of copyright. Works in the *public domain* [38] have had their copyrights expire (after a legally dictated time period), were never eligible for copyright protection due to specific carve-outs (e.g., government documents in the U.S. [2]), or were otherwise dedicated to the public domain by their copyright owners (e.g., with

a CC0 license [35]). Copyright owners can *license* their protected works, allowing others to adapt and reuse them under specified terms. For example, Creative Commons (CC) Licenses (except CC0) grant the right to "reproduce and Share the Licensed Material, in whole or in part; and produce, reproduce, and Share Adapted Material" [36]. For a more in-depth and accessible discussion about licenses and generative AI, see Lee et al. [97, Parts II.I–II.J].

For the Common Pile, we collect and curate public domain and openly licensed text, where we consider "openly licensed" to mean any license that meets the Open Knowledge Foundation's Open Definition 2.1. Some prominent examples of licenses that are considered to be "open" under this definition include CC BY [37], CC BY-SA [39], and software licenses certified by the Blue Oak Council (e.g., the MIT license) [20]. We note that CC NC (non-commercial) and ND (no derivatives) licenses are not considered open under this definition and we therefore do not include content distributed under these licenses. While the use of an open license does not necessarily imply that the rights holder has specifically contemplated use of their content to train LLMs, most open licenses include text like "the above rights may be exercised in all media and formats whether now known or hereafter devised" [37]. Overall, we consider our use of openly licensed data to be a substantial first step towards ethical pre-training dataset curation.

## 2.1 License due diligence

**License laundering** There is a large quantity of data on the internet with incorrect, ambiguous, or missing licensing metadata [97, 107]. A common pitfall is "license laundering," where a copyrighted work is redistributed (typically by a non-rights holder) with an incorrect license. License laundering can undermine our ability to confidently source openly licensed content since it implies that we cannot always trust the license distributed with a piece of content. To address this issue, we set strict standards for data sourcing, only including data from sources where we were confident that the licensing information was provided by the copyright holder, which ultimately led us to exclude certain sources such as OpenAlex [81, 127], YouTube Commons [75], and the Hacker News dataset on Kaggle.

**Use of collection licenses** A related issue is the licensing status of compilations of existing works. Many training corpora are released under open licenses, but these licenses do not necessarily align with the licensing status of the underlying documents [97, Part II.A]. As an example, the ODC-By license has been commonly used for large-scale web corpora such as Dolma [168], FineWeb [133], and TxT360 [174]. ODC-By, by definition, does not extend to *individual documents* within the corpus; therefore, the copyright of documents in these collections is still controlled by the document authors, and does *not* imply that the text itself is openly licensed.

**LLM-generated synthetic datasets** Datasets containing text generated by LLMs trained on unlicensed data have been released under open licenses [e.g. 196]. It has not yet been established whether it is permissible to apply arbitrary licenses to the generations of an LLM that was trained on unlicensed data [97]. We therefore take a conservative stance and avoid synthetic content that was generated by an LLM.

**Caveats and considerations** Despite our best efforts at due diligence, data that falls outside of our curatorial principles and choices may have still ended up in our dataset. License laundering is a notoriously hard problem to identify exhaustively in practice [97]. Further, some *documents* that we collect that are in the public domain or are openly licensed may contain material with unclear status (e.g., quoted snippets of in-copyright books in public domain U.S. government publications).

Copyright owners may also change the license they associate with their content. Since we collected and curated the Common Pile v0.1 in late 2024, the licensing information we include may not be completely aligned with more recent updates. However, with respect to distributing the Common Pile v0.1, such changes in underlying licenses are not a relevant concern. For the licenses we consider, when one obtains a piece of content under a specific license, they acquire it subject to the terms of that license at the time of acquisition. For example, Creative Commons licenses come with specific conditions that apply *perpetually* once the license is granted. The Creative Commons (CC) FAQ states that "CC licenses are not revocable...[A creator] may stop distributing under the CC license at any time, but anyone who has access to a copy of the material [(as is the case for our curation of the Common Pile)] may continue to redistribute it under the CC license terms" [42]. As a result, the license terms under which we obtained data for the Common Pile remain valid; we can freely redistribute this content to others.

Finally, we note that while it is relatively straightforward to obey attribution requirements when redistributing data, attributing model predictions back to the training data points that influenced them remains an active area of research [130, 28].

## 2.2 Comparisons with related work

Our work is not the first that aims to construct a dataset of openly licensed and/or public domain data for the purposes of training machine learning models. Past efforts include CommonCanvas [63], a collection of approximately 70 million Creative Commons-licensed images designed for training image generation models, the PG19 dataset [141] of public domain novels sourced from Project Gutenberg used for benchmarking language models, the C4Corpus tools for sourcing Creative Commons text from Common Crawl snapshots [69], and many datasets comprising CC BY-SA-licensed text from Wikipedia [67, 116].

More relevant to our work are the recent Open License Corpus (OLC) [120], Common Corpus [75, 92], and KL3M [79] datasets, which were constructed for use as LLM pre-training data. On the whole, OLC uses similar selection criteria to ours, including text that is in the public domain or is openly licensed. However, OLC also includes conversations scraped from Hacker News, which does not have an open license. Additionally, OLC is considerably smaller than the Common Pile v0.1, comprising data from 12 sources (vs. 30 for Common Pile) totaling 0.85 TB of text (vs. 7.6 TB for Common Pile). Common Corpus also uses a similar set of allowable licenses/copyright statuses (e.g. CC BY, CC BY-SA, public domain, MIT-style, etc.) although the specific licenses/statuses are not clear because Common Corpus does not retain full per-document licensing information across all sources. Additionally, Common Corpus incorporates data from OpenAlex [127] which is known to provide inaccurate licensing information [e.g., 81]. Furthermore, while the Common Pile and Common Corpus are similar in size (7.6 TB vs. 7.4 TB), Common Corpus targets a broader set of languages and therefore contains significantly less English text. Conversely, KL3M does not consider CC BY-SA to be acceptable and, as a result, almost exclusively consists of government documents. Accordingly, the Common Pile is much larger than KL3M (3 TB), and is built from significantly more diverse data sources (Figure 1 & Section 3). In subsection 4.3, we compare the Common Pile v0.1 to these datasets in a controlled setting, ultimately showing that it produces substantially more performant LLMs.

## 3 Composition of the Common Pile

The Common Pile comprises content drawn from a wide range of domains, including scholarly publications, government documents, online discussions, books, open educational resources, and more. In this section, we provide an overview of each of the domains contained in the Common Pile and briefly discuss their constituent data sources. In-depth discussion of each source is provided in Appendix B.

**Scientific and scholarly texts**, which are often distributed under open licenses due to open access mandates, appear in many LLM pre-training datasets [e.g. 58, 168, 186] since they expose models to technical terminology, formal reasoning, and long-range document structure. To attain broad coverage of scholarly text, we filter peS2o [167] (a collection text extracted from open-access scientific PDFs based on S2ORC [105]) to only retain openly licensed research papers. For medical-domain text, we collect text from openly licensed articles in the U.S. National Institutes of Health's National Library of Medicine's PubMed Central archive. Additionally, we collect data from ArXiv, which contains over 2.4 million articles in the quantitative sciences, most of which are uploaded as LaTeX source and may be distributed under various licenses chosen by a given article's author. We include openly licensed articles sourced from ArXiv's bulk-access S3 bucket and parsed using LaTeXML and Trafilatura [10]. Furthermore, according to ArXiv's licensing policy, all metadata (including abstracts) of articles posted to ArXiv are distributed under the CC0 license; we therefore include the abstracts for *all* ArXiv papers in the Common Pile, regardless of the paper's full-text license.

**Online discussion forums** comprise multi-turn question-answer pairs and discussions and therefore can be useful for training language models to follow conversational structure as well as for improving performance on question answering and dialogue-centric tasks. StackExchange is a collection of websites that host user-provided questions and answers and allow their redistribution under a CC BY-SA license. We leverage the user-provided StackExchange dumps from the Internet Archive and format questions/answers in the same order they appear on StackExchange, using PyMarkdown to

convert each comment into plain text. Additionally, we collect text from issues, pull requests, and comments on GitHub, which, according to GitHub's terms of service, inherit the license of their associated repository. We extract this content from repositories with Blue Oak Council-approved licenses from the GitHub Archive. Finally, we include logs of all discussions on the Ubuntu-hosted Internet Relay Chat (IRC) since 2004, which are released into public domain.

**Government and legal texts** are often published directly into the public domain or under open licenses. For example, in the US, text written by federal government employees is considered to be in the public domain. We therefore include all plain-text documents made available through the United States Government Publishing Office (USGPO)'s GovInfo.gov developer API. Additionally, we include all plain text regulatory documents published by U.S. federal agencies from Regulations.gov, an online platform that hosts newly proposed rules and regulations from federal agencies. The Common Pile also incorporates US Patents and Trademark Office (USPTO) patent documents sourced from the Google Patents Public Data dataset [78], containing millions of public domain patents and published patent applications dating back to 1782. Similarly, the Hansard (the official record of parliamentary proceedings) of the United Kingdom is distributed under the Open Parliament License, which stipulates similar terms to the CC BY license. We source UK Hansard data from ParlParse [132], covering Commons debates from 1918 forward and Lords proceedings from the 1999 reform. For legal text, we leverage the Caselaw Access Project (comprising 40 million pages of U.S. federal and state court decisions and judges' opinions from the last 365 years) and Court Listener (including 900 thousand cases scraped from 479 courts). Only legal texts in the public domain were selected for the Common Pile.

**Curated task datasets** are typically designed for fine-tuning on specific downstream tasks such as question answering, summarization, or text classification. To source datasets that are distributed under an open license and only contain content owned by the dataset's rights holder (to avoid license laundering), we use metadata and redistributed datasets from the Data Provenance Initiative [107, 110]. Full details on the datasets we include are available in Appendix D.

**Books**, particularly historic text, can fall into the public domain due to copyright expiration—for example, in the United States, books published prior to 1929 are currently in the public domain. We source public domain books from various sources, including the Biodiversity Heritage Library (BHL), an open-access digital library for biodiversity literature and archives; pre-1929 books digitized by the Internet Archive on behalf of HathiTrust member libraries; the collection of public domain books called "Selected Digitized Books" released by the Library of Congress; and select books from Project Gutenberg, an online collection of over 75,000 digitized books, most of which are in the public domain.

**Open Educational Resources (OERs)** are educational materials (e.g. textbooks, lecture notes, lesson plans, etc.), typically published under Creative Commons licenses that support free and equitable access to education. We collect data from multiple OER repositories, including the Directory of Open Access Books (DOAB), an online index of over 94,000 peer-reviewed books curated from trusted open-access publishers; PressBooks, a searchable catalog of over 8,000 open access books; OERCommons, an online platform where educators share open-access instructional materials; and LibreTexts, a catalog of over 3,000 open-access textbooks.

**Wikis** are topic- or domain-specific encyclopedic websites that are collaboratively written, maintained, and moderated. Historical and cultural precedent has led many wikis to have an open license. We downloaded the official database dumps of wikitext (Mediawiki's custom markup language) of the English-language wikis that are directly managed by the Wikimedia foundation and converted wikitext to plain text using `wtf_wikipedia`. For wikis not managed by Wikimedia, we make use of wikiteam's unofficial database dumps and apply the same conversion process.

**Source code** has proven to be a useful part of LLM pre-training corpora, not only to support coding abilities but also to improve reasoning [7, 114, 121]. Due to the Free and Open Source Software (FOSS) movement, a great deal of source code is distributed with an open license. We leverage prior work done by the Software Heritage Foundation and BigCode to compile the openly licensed subset of the Stack V2 [114], based on the license detection performed by the creators of Stack V2. Additionally we collected all Python Enhancement Proposals (PEPs)—design documents that generally provide a technical specification and rationale for new features of the Python programming language—that were released into the public domain.

**YouTube** allows users to upload content under a CC BY license. We therefore sourced and transcribed speech-heavy CC BY videos from YouTube. To avoid license laundering and focus on high-quality speech-based textual content, we manually curated a set of over 2,000 YouTube channels that release original openly licensed content containing speech. From these channels, we retrieved and transcribed (using Whisper [139]) over 1.1 million openly licensed videos comprising more than 470,000 hours of content.

**Web text** is a common source of LLM pre-training data. A small fraction of content on the web is distributed under open licenses. To recover a portion of this content, we process 52 Common Crawl snapshots using a regular expression (regex) adapted from the C4Corpus project [69] to retain pages that include a CC BY, CC BY-SA, or CC0 marker. This regex naturally results in many false positives (e.g., it would retain a page that included and provided attribution for a CC BY image but otherwise contained unlicensed content), so we manually verified the top 1000 domains by content volume, retaining only those for which all content was assigned a Creative Commons license. Text was extracted using a pipeline similar to the one used for Dolma [168]. We provide more details on the composition of our web-sourced text, called CCCC, in Appendix G. Apart from CCCC, we additionally manually collected data from a few select sites, including Foodista, a community-maintained site with recipes and food-related news as well as nutrition information; news sites that publish content under CC BY or CC BY-SA according to Open Newswire; and the Public Domain Review, an online journal dedicated to exploration of works of art and literature that have aged into the public domain.

## 4 Assessing the Common Pile v0.1's quality

The utility of an LLM pre-training dataset is mostly assessed in terms of whether or not it can be used to train performant LLMs. To validate our efforts in curating the Common Pile, we use it as the basis of an LLM pre-training dataset created through additional filtering (subsection 4.1) and rebalancing (subsection 4.2). Then, we perform a controlled data ablation study (subsection 4.3) where we train otherwise-identical LLMs on different pre-training datasets, including prior datasets comprised of openly licensed text mentioned in subsection 2.2 as well as a selection of representative pre-training datasets of unlicensed text. Finally, we train Comma v0.1-1T and Comma v0.1-2T, 7 billion parameter LLMs trained on 1 and 2 trillion tokens (respectively) of Common Pile-sourced content, and compare them to models with a similar parameter count and training budget that were trained on unlicensed text (subsection 4.4).

### 4.1 Dataset preprocessing and filtering

Before training a language model, it is considered important to "clean" data in hopes of retaining only high-quality text under some notion of quality [4, 106]. Consequently, before training on data from the Common Pile (which is distributed in a relatively "raw" format), we independently preprocessed each of the Common Pile's non-code datasets using pipelines implemented with the Dolma data processing toolkit [168].

Since the Common Pile v0.1 focuses primarily on English content, we apply **language identification** using a FastText classifier [82] to filter out non-English text. When processing web text from CCCC, we employ the **text quality classifier** adapted from DataComp-LM [100] with an extremely low threshold to remove noisy text. We remove documents with pervasive OCR errors using the **likelihood-based filtering** approach from [167], which removes documents that are assigned an excessively low log-likelihood under a unigram language model constructed from the Trillion Word Corpus [118]. To reduce the prevalence of toxic or inappropriate content, we apply a pair of FastText **toxicity classifiers** implemented in Dolma [168] that were trained on the Jigsaw Toxic Comment Classification Challenge dataset [30]. We apply regex-based **personally identifiable information (PII) redaction** to remove email addresses, phone numbers, and IP addresses, and replace them with <EMAIL_ADDRESS>, <PHONE_NUMBER>, and <IP_ADDRESS> respectively. Finally, we perform source-specific **regex filtering** to remove repetitive or boilerplate text (e.g., page numbers, document preambles, license statements, etc.). For a detailed breakdown of the pre-processing applied to each dataset, see Table 5 (appendix).

After filtering, we perform global document-level fuzzy deduplication across all sources, as excessive data duplication is known to harm language modeling performance [95] and increase memorization

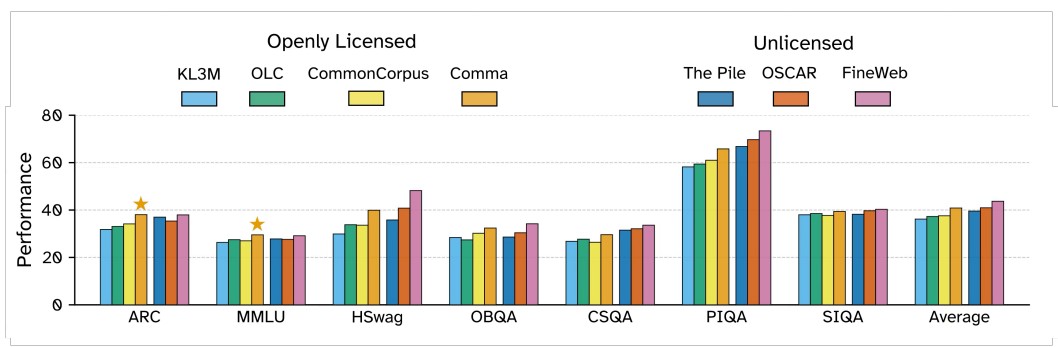

Figure 2: **The Common Pile consistently outperforms other openly licensed corpora as a pre-training dataset.** Following the setup from Penedo et al. [133], we train and evaluate 1.7B parameter models on 28B tokens of data from each dataset. Stars denote benchmarks on which the model trained using the Common Pile outperforms all other models.

[84]. We use the bloom filter-based deduplication functionality from Dolma [168] and deem two documents duplicates if they share more than 90% of their 20-grams.

For code data from the Stack v2, we apply the Red Pajama V1 [186] code filtering heuristics. These include filters based on the mean and maximum line length in a document, the proportion of alphanumeric characters, and the ratio of alphabetical characters to tokens. After this initial filter, we adopt the process used by SmolLM2 [5] where we keep only code in Python, C, C++, SQL, Java, PHP, Rust, Javascript, Typescript, Go, Ruby, Markdown, C#, Swift, or shell and filter this set using language-specific quality classifiers to retain only educational and well-documented code. We use a lower threshold to filter out low-quality code than was used for SmolLM2, resulting in a larger set of post-filtered text. Finally, we extract plaintext from HTML documents in the Stack V2 using Trafilatura [10] and apply our standard plaintext filtering pipeline including language, length, toxicity, and PII filtering.

## 4.2 Data mixing

Recent work [3, 179, 190] has shown that up- or down-weighting pre-training data sources in accordance with some notion of data quality can produce more performant models. Indeed, the sources in the Common Pile vary drastically in their characteristics, and we don't necessarily expect that our largest sources contain the highest quality text. For example, patent text sourced from the USPTO (our second-largest source) exhibits substantially different wording, terminology, and repetition than typical natural language. Consequently, we anticipate that appropriately mixing the sources in the Common Pile (rather than simply combining all sources, i.e., mixing in proportion to source size) is of particular importance. Additionally, while LLM pre-training datasets have been continuously scaled to avoid the diminishing returns that result from repeating data [95], recent work has highlighted that repeating high-quality data can be preferable to avoiding repetition by training on low-quality data [121, 53].

To determine mixing weights, we first trained per-source language models using the procedure outlined in subsection 4.3 below for 28 billion tokens on all sources that were sufficiently large to be repeated less than four times at this data budget. Based on the performance of these per-source models, we heuristically set mixing weights to up- and down-weight high- and low-performance sources respectively while targeting a maximum of six repetitions over the course of a 1 trillion token training run. Additionally, we assumed that our smaller sources were high quality and set their mixing rates such that they were also repeated six times over the course of 1 trillion tokens. The resulting mixture and per-source repetition rates are given in Table 7 (appendix). We also experimented with using MixMin [179] to automatically determine mixing weights but found that it did not improve over our heuristically determined mixture.

Because we use this dataset mixture to train the Comma v0.1 models (subsection 4.4) and because it comprises a heavily filtered and remixed version of the Common Pile v0.1, we refer to it as the "Comma dataset" to distinguish it from the Common Pile itself.

## 4.3 Controlled dataset quality experiments

As a preliminary measure of the Common Pile's quality, we adopt the experimental setting of Penedo et al. [133] and identically train models on the Comma dataset and various preexisting datasets. By using a controlled setting across datasets, we can assert that differences in model performance stem primarily from the quality of each dataset. Specifically, we train 1.7 billion parameter decoder-only Transformer models [183] that follow the Llama architecture [180] on 28 billion tokens of data from each dataset, tokenized using the GPT-2 tokenizer [138]. We follow the hyperparameters and setup of Penedo et al. [133] exactly, except that we used a weight decay of 0.2 instead of 0.1 due to slightly improved performance (possibly due to the large amount of repetition in the Comma dataset).

Each model was then evaluated using the set of "early signal" tasks identified by Penedo et al. [133] which cover commonsense reasoning and knowledge capabilities; specifically, we evaluate zero-shot performance on ARC [33], MMLU [71], HellaSwag (HSwag) [193], OpenBookQA (OBQA) [119], CommonSenseQA (CSQA) [172], PIQA [19], and SocialIQA (SIQA) [161]. We omit Winogrande (which was used in Penedo et al. [133]) because it is included in the set of datasets we sourced from the Data Provenance Initiative; consequently all of the tasks we evaluate on are "unseen" by all models. We highlight that a significant portion of the Comma dataset is code, but none of the tasks we evaluate on measure code capabilities. While it is possible that we could improve performance by omitting code data in this setting, we retained code for reliable reporting of the Comma dataset's performance.

As baselines, we compare to the prior datasets that aim to provide open licensed text discussed in subsection 2.2: OLC [120], Common Corpus [92], and KL3M [79]. We additionally compare to the Pile, as it one of the only LLM pre-training datasets that contains a comparable number of diverse sources to the Common Pile (22 vs. 30). Finally, we report the performance of two web text-based unlicensed pre-training datasets: OSCAR [170], which incorporates relatively little filtering; and FineWeb [133], an recent dataset that reflects current best practice for LLM pre-training dataset curation.

The resulting performance of each model is shown in Figure 2, with detailed results in Table 9 (appendix). Notably, the Comma dataset-based model outperforms the models trained OLC, Common Corpus, and KL3M across all benchmarks and outperforms the Pile-based model on all but two benchmarks. While the performance of the FineWeb-based model is the best on most benchmarks, the Comma dataset-based model performs best on the scientific and scholarly knowledge-based benchmarks MMLU and ARC, possibly due to the Common Pile's large proportion of domain-relevant text. On the other hand, on the commonsense reasoning datasets HellaSwag, PIQA, and CommonSenseQA, the model trained on the Comma dataset has significantly worse performance than models trained on the Pile, OSCAR, and FineWeb, possibly indicating a lack of relevant data in the Common Pile. We additionally note that recent work [189] highlights that performance on HellaSwag is most heavily influenced by coverage of certain domains and topics such as personal blogs, tutorials, hobbies, and sports, which are poorly represented in the Common Pile. Overall, these findings confirm that the Comma dataset performs best among datasets that aim to contain only openly licensed data and is also a strong candidate in general, particularly when targeting scientific and scholarly applications.

We note that the Comma dataset is the only dataset we evaluate that explicitly includes task-like data due to inclusion of data from the Data Provenance Initiative (DPI). To verify that this does not confer an unfair advantage, we trained an additional model on the Comma dataset with all sources retained except for the DPI-sourced data. Removing this source had a minimal impact on model performance (full results in Table 9), with a notable decrease only on HellaSwag, possibly suggesting that the DPI data contains domain- or task-relevant data for this benchmark that our other sources lack.

## 4.4 Comma v0.1

Having established that Comma's dataset produces models with competitive performance when compared to other datasets, we now validate our efforts at larger, more realistic scales. We train Comma v0.1-1T and Comma v0.1-2T, a pair of 7 billion-parameter LLMs trained on 1 and 2 trillion tokens of text respectively, and compare with other models trained using similar computational budgets.

**Tokenization**  While training a tokenizer on unlicensed text is less likely to raise ethical or IP-related issues than training an LLM, we nevertheless trained a custom tokenizer on the Comma dataset to

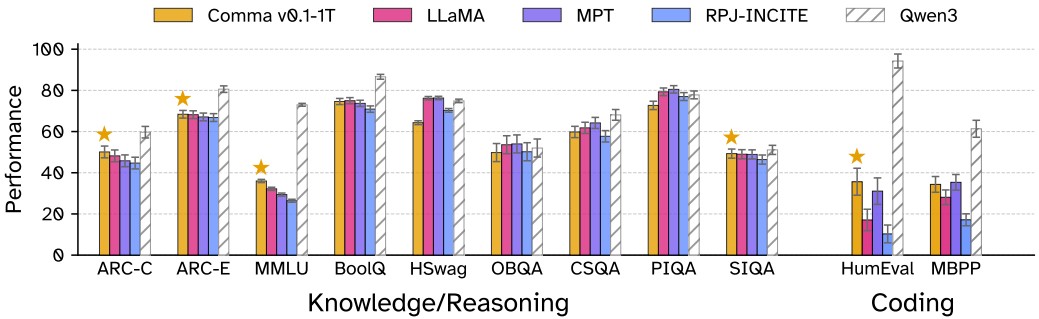

Figure 3: **Comma v0.1-1T performs comparably to models trained with similar resources (7 billion parameters, 1 trillion tokens) on several standard benchmarks.** To contextualize these results, we include Qwen3 8B (trained on 36 trillion tokens) as a "current best-practices" upper bound. Stars denote benchmarks on which Comma v0.1-1T achieves the highest score among compute-matched models (i.e., all models other than Qwen3). Error bars represent 95% confidence intervals. Full numerical results are provided in Table 10 (appendix).

ensure that our entire modeling pipeline was based on openly licensed data. In addition, the different characteristics of our dataset likely makes existing tokenizers (which are often trained on web text) suboptimal. We therefore trained a BPE-based [56] tokenizer using the Hugging Face `tokenizers` library using a vocabulary size of 64,000. We follow the same splitting regex as Llama 3.2 [64] and the Hugging Face `ByteLevel` preprocessor; no Unicode normalization was used. The tokenizer was trained on a 600GB sample [151] of text from the Comma dataset.

**Training setup**     We trained Comma v0.1-1T and -2T using the `lingua` framework [184]. We base our model architecture and training hyperparameters on `lingua`'s Llama-7B configuration, which closely follows the conventions set by the Llama series of models [180, 64]. For Comma v0.1-1T, we trained with an effective batch size of 2M tokens (512 sequences with 4096 tokens each), using the AdamW [112] optimizer and a weight decay of 0.2. For the Comma v0.1 variant trained on 2 trillion tokens, we increased the batch size to 8.3M tokens (2048 sequences with 4096 tokens each).

We performed two stage training, with a first stage following a cosine learning rate schedule and the second stage being a 'cool-down' [74], where we train only on a subset of high-quality sources using the mixing weights provided in Table 8 (appendix) while decaying the learning rate linearly to 0. Comma v0.1-1T had a first stage of 460,000 steps with 2,000 steps of warmup, an initial learning rate of $1e-3$, a minimum learning rate of $1e-9$, a cosine schedule period of 500,000 steps, and 18,000 steps of decay. For Comma v0.1-2T, the first stage instead had 230,000 steps with a maximum and minimum learning rate of $2e-3$ and $2e-9$ respectively and a period of 250,000 steps, with 9,000 steps of decay in the second stage. For both models, we average together ten evenly spaced checkpoints from the cool-down phase to produce a final model as suggested by Grattafiori et al. [64]. Apart from our main Comma v0.1-1T and -2T training runs, we completed several additional runs to better understand how hyper-parameters impact the model, including using a different batch size and following a three-stage (rather than two-stage) curriculum. Overall, the results of these runs were consistent with the findings from our main training runs. Additional details can be found in Appendix O.

**Evaluation**     We evaluate the Comma v0.1 models following the evaluation setup of Groeneveld et al. [66], supplemented with two code benchmarks. Specifically, we evaluate models on ARC [33], MMLU [71], BoolQ [31], HellaSwag [193], OpenBookQA [119], CommonsenseQA [172], PIQA [19] and SIQA [161] to probe world knowledge and reasoning, and HumanEval [25] and MBPP [8] to evaluate coding capabilities. All evaluations use the Language Model Evaluation Harness [59] with zero-shot prompting. For the coding tasks, we report pass@10 over 20 attempts.

**Baseline models**     As our goal is to demonstrate that models trained on our dataset perform similarly to ones trained on unlicensed data, we primarily compare to prior models with the same parameter count and token budget. As far as we are aware, no other models trained on openly licensed data exist at this scale. For Comma v0.1-1T, we compare to Llama 1 7B [180], MPT-7B [176], RPJ-INCITE-7B [186], StableLM-7B [12], and OpenLLaMA-7B [60]. For the two trillion token variant, we compare to OLMo Twin (specifically OLMo-7B-Twin-2T) [66], Llama 2 7B [181], and DeepSeekLLM [16]. Over time, the token budgets of open pre-trained LLMs have continually

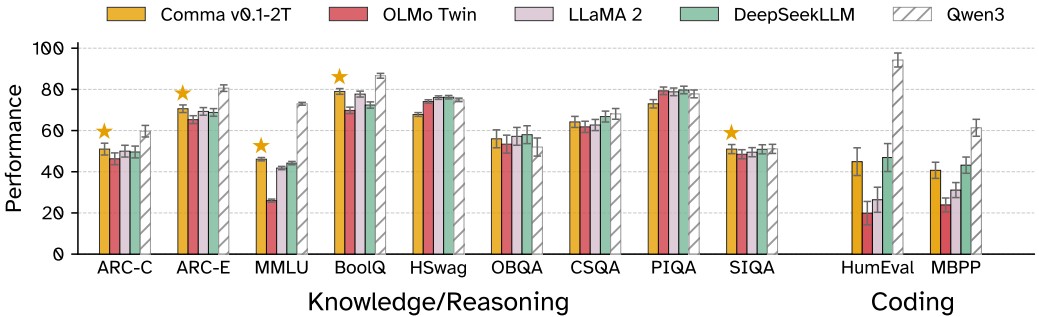

Figure 4: **Comma v0.1-2T is also competitive with budget-matched models (7 billion parameters, 2 trillion tokens) trained on unlicensed data.** We additionally include Qwen3 8B as a higher budget upper bound. Stars denote benchmarks where Comma v0.1-2T achieves the highest score among compute-matched models. Full numerical results are provided in Table 11 (appendix).

grown [144], and current standard practice is to pretrain on significantly more than 1 or 2 trillion tokens. Consequently, recent models tend to outperform our baselines, which were released in 2023 and 2024. To provide a state-of-the-art point of reference, we additionally include results for the recently released Qwen3 8B [177], which was trained for 36 trillion tokens. We emphasize that we cannot reliably compare to a model with a 36× or 18× larger training budget and we primarily include it as a point of reference.

**Results** As shown in Figure 3, Comma v0.1-1T performs comparably to budget-matched baseline models on several of the benchmarks tested. In line with our results from subsection 4.3, we observe that Comma v0.1-1T excels on knowledge-based benchmarks like MMLU, but lags behind on HellaSwag and PIQA. Comma v0.1-1T is also particularly strong at code-related tasks where it outperforms most baseline models by a wide margin. Comparisons to StableLM and OpenLLama can be found in Table 10 (appendix), but show similar trends.

Our promising results from training on 1 trillion tokens motivated us to experiment with longer training durations. To test whether the filtered dataset supports training durations beyond 1T, we trained Comma v0.1-2T simply by repeating the same data mixture used for Comma v0.1-1T approximately twice. We note that this pre-training mixture involves repeating certain sources an excessive number of times (up to 16 passes for some sources). Prior work suggests that these extreme levels of data repetition may result in diminishing returns [122]. However, these experiments still give us a preliminary picture of the performance achievable under a larger budget.

The performance of Comma v0.1-2T is compared with budget-matched models in Figure 4. Notably, we find that Comma v0.1-2T is competitive with OLMo, Llama 2, and DeepSeekLLM, with especially strong performance on MMLU, SIQA, ARC-E, and the coding tasks. We emphasize that the Comma v0.1-2T result here is likely *not* a best-case 2T-token run using the Common Pile v0.1 due to excessive repetition, and better performance could likely be attained through a 2T-specific mixture and curriculum. Nevertheless, this result highlights the promise of larger scale training runs based on the Common Pile. Qwen3 8B's superior performance across most benchmarks confirms the benefit of larger training budgets and motivates future efforts on scaling up the Common Pile.

## 5 Conclusion

We release *Common Pile v0.1*, an 8TB corpus that—to our knowledge—constitutes the largest dataset built exclusively from openly licensed text. Alongside our dataset, we release *Comma v0.1-1T and -2T*, two performant 7-billion-parameter LLMs trained on text from the Common Pile, as well as the filtered and rebalanced data mixture we used for training. Our results demonstrate that not only is the Common Pile the strongest dataset for pretraining under an open-license constraint, but also that it produces models comparable to those trained on an equivalent amount of unlicensed data. This positive result holds promise for future of open-license pretraining, especially if the research community invests in collecting larger quantities of openly licensed text data in the future. Ultimately, we believe that the Common Pile v0.1 represents the first step on the path towards a more ethical language model ecosystem, where performance need not come at the cost of creator rights and legal transparency.

## Acknowledgments

We thank Chris Maddison, Anvith Thudi, Pierre-Carl Langlais, Alec Radford, Adam Roberts, Sewon Min, Weijia Shi, and Jonas Geiping for fruitful discussions and constructive feedback. We also thank Nikoli Dryden and the LLNL LBANN team for contributing scalable model training tools to the open-source community that were useful in supporting experiments on LLNL's cluster. An early draft of this work was shared at the Dataset Convening hosted by the Mozilla Foundation and EleutherAI. We thank the participants for their discussion and feedback.

This work was supported by funding from the Mozilla Foundation and Sutter Hill Ventures. We acknowledge the support of the Natural Sciences and Engineering Research Council of Canada (NSERC). Researchers funded through the NSERC-CSE Research Communities Grants do not represent the Communications Security Establishment Canada or the Government of Canada. Any research, opinions or positions they produce as part of this initiative do not represent the official views of the Government of Canada.

Parts of this work were performed under the auspices of the U.S. Department of Energy by Lawrence Livermore National Laboratory under Contract DE-AC52-07NA27344 and was supported by the LLNL-LDRD Program under Project No. 24-ERD-010 and Project No. 24-SI-008 (LLNL-CONF-2006420).

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

# Appendix

## Table of Contents

## A  Contributions

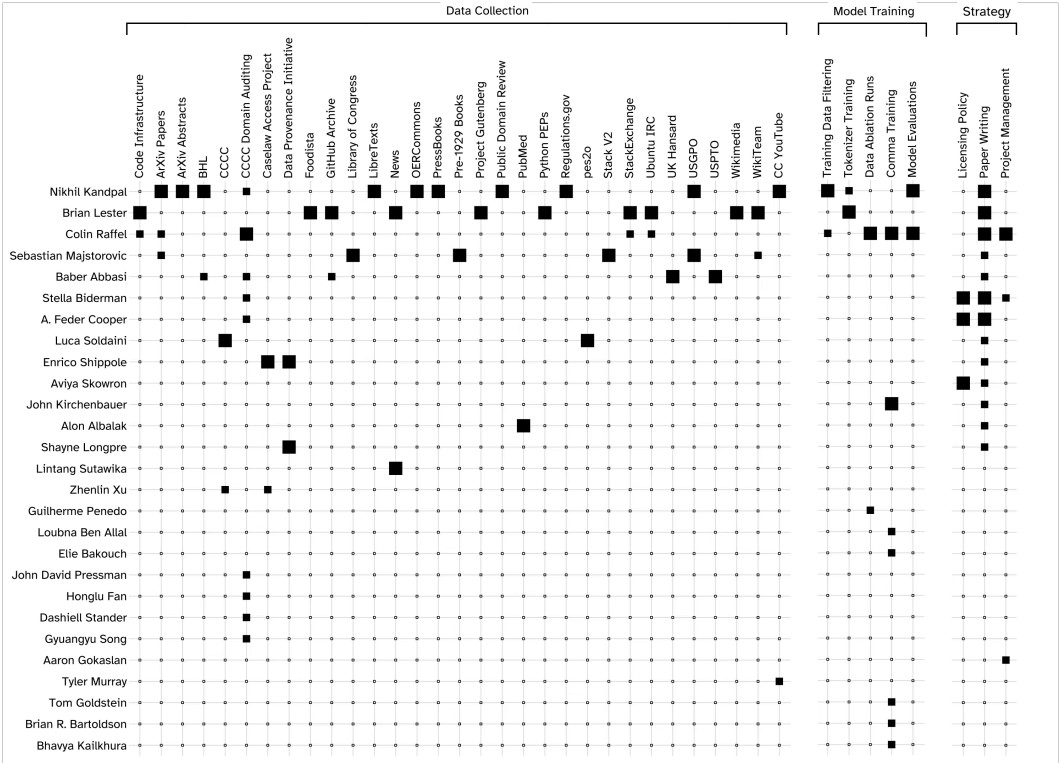

Figure 5: **Author contributions to this work.** Large squares indicate a major contribution and small squares indicate a supporting contribution.

## B  Detailed Description of Sources

Below, we give a more in-depth overview of the sources that make up the Common Pile, including specific license decisions and tools used during collection.

### B.1  Scientific and Scholarly Text

Scientific and scholarly texts are a staple of modern LLM pretraining corpora, appearing in nearly all large-scale datasets [e.g. 58, 186, 168] since they expose models to technical terminology, formal reasoning, and long-range document structure—skills that are essential for downstream tasks in science, education, and question answering. Thanks to open access mandates and academic cultural norms, many scholarly texts are either in the public domain or are distributed under open licenses.

**peS2o**  To ensure broad coverage across many scientific disciplines, we include a version of peS2o [167] restricted to openly licensed articles. pes2o is derived from S2ORC [105], a corpus of openly licensed abstract and full-text papers that have been converted to a structured format using Grobid [65]. Starting from Grobid's XML output, peS2o filters papers that are too short, have incorrect metadata, are in languages other than English, and contain OCR errors using a combination of heuristic- and model-based filtering steps. We refer the reader to the datasheet and code for more details on this processing pipeline. The subset of peS2o included in the Common Pile starts from v3 of the corpus, which contains documents from January 1, 1970 to October 6, 2024. We retain full-text papers with CC BY, CC BY-SA, or CC0 licenses, or that have been labeled as public domain; metadata is provided by the Semantic Scholar APIs [88]. After filtering, this set contains 6.3 million

papers, or 35.7 billion whitespace-separated segments. We provide more details on the composition of this subset in Appendix H.

**PubMed**  PubMed Central (PMC) is an open-access archive of biomedical and life sciences research papers maintained by the U.S. National Institutes of Health's National Library of Medicine. We collected papers from PMC whose metadata indicated that the publishing journal had designated a CC BY, CC BY-SA, or CC0 license. PMC stores the text content of each article as a single XML file, which we convert to markdown using pandoc.

**ArXiv Papers**  ArXiv is an online open-access repository of over 2.4 million scholarly papers covering fields such as computer science, mathematics, physics, quantitative biology, economics, and more. When uploading papers, authors can choose from a variety of licenses. We included text from all papers uploaded under CC BY, CC BY-SA, and CC0 licenses in the Common Pile through a three-step pipeline: first, the latex source files for openly licensed papers were downloaded from ArXiv's bulk-access S3 bucket; next, the LaTeXML conversion tool was used to convert these source files into a single HTML document; finally, the HTML was converted to plaintext using the Trafilatura [10] HTML-processing library.

**ArXiv Abstracts**  Each paper uploaded to ArXiv includes structured metadata fields, including an abstract summarizing the paper's findings and contributions. According to ArXiv's licensing policy, the metadata for any paper submitted to ArXiv is distributed under the CC0 license, regardless of the license of the paper itself. Thus, we include as an additional source the abstracts for every paper submitted to ArXiv. We source the abstracts from ArXiv's API via the Open Archives Initiative Protocol for Metadata Harvesting endpoint and reproduce them as-is.

## B.2  Online Discussions and Forums

Online forums are a rich source of multi-turn, user-generated dialogue covering a wide range of topics. These platforms often feature question–answer pairs, problem-solving discussions, and informal explanations of technical and non-technical concepts. The Common Pile incorporates online discussions from sources that distribute content under an open license.

**StackExchange**  While StackExchange formerly provided structured XML dumps of all of their content, since July of 2024, StackExchange has stopped publishing dumps to the Internet Archive. Instead, each site can provide a logged-in user with a custom URL to download the dump for that site. This means that dumps for defunct sites like windowsphone.stackexchange.com are inaccessible. Additionally, in dumps produced by the new export tool, many questions that are available in past dumps (and accessible on the site) are not present. We therefore extract all questions and answers from community uploaded dumps from December of 2024 from the Internet Archive and additionally extract missing questions and answers from the last official dumps in July of 2024 to account for the deficiencies listed above. We use a question, its comments, its answers and the comments on each answer as a single document. Following the display order on StackExchange, answers are ordered by the number of votes they received, with the exception that the "accepted answer" always appears first. PyMarkdown was used to convert each comment into plain text.

**GitHub Archive**  According to GitHub's terms of service, issues and pull request descriptions—along with their comments—inherit the license of their associated repository. To collect this data, we used the GitHub Archive's public BigQuery table of events to extracted all issue, pull request, and comment events since 2011 and aggregated them into threads. The table does not include "edit" events so the text from each comment is the original from when it was first posted. We filtered out comments from bots. This resulted in approximately 177 million threads across 19 million repositories. We then removed threads whose repositories did not have a Blue Oak Council-approved license. License information for each repository comes from either 1) the "public-data:github_repos" BigQuery Table, 2) metadata from the StackV2, or 3) the GitHub API. License filtering left 10 million repositories. PyMarkdown was used to convert from GitHub-flavored markdown to plain text. When parsing failed, the raw markdown was kept.

**Ubuntu IRC**  Logs of all discussions on the Ubuntu-hosted Internet Relay Chat (IRC) since 2004 have been archived and released into the Public Domain. We downloaded all chats from all channels up until March of 2025. We consider all messages for given channel on a given day as a single document. We removed system messages as well as those from known bots.

### B.3 Government and Legal Texts

Governments produce a vast amount of informational text, ranging from legislation and legal opinions to scientific reports, public communications, and regulatory notices. This content is explicitly intended to inform the public, and as such, in many jurisdictions it is published directly into the public domain or under open licenses. In the United States, for example, works authored by federal employees as part of their official duties are not subject to copyright. Government and legal texts offer language models exposure to formal argumentation, legal reasoning, and procedural language.

**US Government Publishing Office**    The United States Government Publishing Office (USGPO) is a federal agency responsible for disseminating official documents authored by the U.S. government. The Common Pile v0.1 includes all plain-text documents made available through the USGPO's GovInfo.gov developer API. This collection comprises over 2.7 million documents, spanning issues of the Federal Register, congressional hearing transcripts, budget reports, economic indicators, and other federal publications.

**US Patents and Trademark Office**    In the US, patent documents are released into the public domain as government works. Patents follow a highly standardized format with distinct required sections for background, detailed description, and claims. We include parents from the US Patents and Trademark Office (USPTO) as provided by the Google Patents Public Data dataset [78], which includes millions of granted patents and published patent applications dating back to 1782. We processed these documents to extract clean text while preserving this structured format. Mathematical expressions and equations were converted into LaTeX.

**Caselaw Access Project and Court Listener**    The Common Pile contains 6.7 million cases from the Caselaw Access Project and Court Listener. The Caselaw Access Project consists of nearly 40 million pages of U.S. federal and state court decisions and judges' opinions from the last 365 years. In addition, Court Listener adds over 900 thousand cases scraped from 479 courts. The Caselaw Access Project and Court Listener source legal data from a wide variety of resources such as the Harvard Law Library, the Law Library of Congress, and the Supreme Court Database. From these sources, we only included documents that were in the public domain. Erroneous OCR errors were further corrected after digitization, and additional post-processing was done to fix formatting and parsing.

**UK Hansard**    Hansard represents the official record of parliamentary proceedings across the United Kingdom's legislative bodies. The Common Pile incorporates records from multiple sources, including debates and written answers from the UK Commons and Lords, devolved legislatures (Scottish Parliament, Senedd in both English and Welsh, Northern Ireland Assembly), London Mayor's Questions, and ministerial statements. Data was sourced from ParlParse [132], covering Commons debates from 1918 forward and Lords proceedings from the 1999 reform. Each document was processed to preserve complete parliamentary sessions as cohesive units, maintaining the natural flow of debate. All content is published under the Open Parliament License [182].

**Regulations.gov**    Regulations.gov is an online platform operated by the U.S. General Services Administration that collates newly proposed rules and regulations from federal agencies along with comments and feedback from the general public. The Common Pile includes all plain-text regulatory documents published by U.S. federal agencies on this platform, acquired via the bulk download interface provided by Regulations.gov.

### B.4 Curated Task Data

Curated datasets that cover specific tasks such as question answering, summarization, or text classification are often released via open licenses to the research community. While not traditionally part of pretraining corpora, including a small amount of task-oriented data during pretraining can help models acquire early familiarity with task formats and prompt–completion structures.

**Data Provenance Initiative**    The Data Provenance Initiative is a digital library of supervised datasets that have been manually annotated with their source and license information [107, 110]. We leverage their tooling to filter HuggingFace datasets, based on a range of criteria, including their licenses, which may be particularly relevant for supervised datasets [115]. Specifically, we filter the data according to these criteria: contains English language or code data, the text is not model-generated, the dataset's audit yielded a open license and the original sources of the data are only from recognized public domain sources.

## B.5 Books in the Public Domain

Books represent a time-tested resource for language model pretraining, offering carefully edited, long-form prose that supports learning of narrative coherence and long-range dependency modeling. For these reasons, many large-scale pretraining corpora—including the Pile [58], Dolma [168], and RedPajama [186]—include content from books [41]. In the United States, as of 2024, books published prior to 1929 are in the public domain. Thus, the Common Pile includes public domain books drawn from curated collections, covering topics such as literature, science, and history.

**Biodiversity Heritage Library** The Biodiversity Heritage Library (BHL) is an open-access digital library for biodiversity literature and archives. The Common Pile contains over 42 million public domain books and documents from the BHL collection. These works were collected using the bulk data download interface provided by the BHL and were filtered based on their associated license metadata. We use the optical character recognition (OCR)-generated text distributed by BHL.

**Pre-1929 Books** Books published in the US before 1929 passed into the public domain on January 1, 2024. We used the bibliographic catalog Hathifiles produced by HathiTrust to identify digitized books which were published in the US before 1929. The collection contains over 130,000 books digitized and processed by the Internet Archive on behalf of HathiTrust member libraries. The OCR plain text files were downloaded directly from the Internet Archive website.

**Library of Congress** The Library of Congress (LoC) curates a collection of public domain books called "Selected Digitized Books". We downloaded over 130,000 English-language books from this public domain collection as OCR plain text files using the LoC APIs.

**Project Gutenberg** Project Gutenberg is an online collection of over 75,000 digitized books available as plain text. We use all books that are 1) English and 2) marked as in the Public Domain according to the provided metadata. Additionally, we include any books that are part of the pg19 [141] dataset, which only includes books that are over 100 years old. Minimal preprocessing is applied to remove the Project Gutenberg header and footers, and many scanned books include preamble information about who digitized them.

## B.6 Open Educational Resources

Open Educational Resources (OERs) are educational materials, typically published under open licenses, to support free and equitable access to education. These resources include educational artifacts such as textbooks, lecture notes, lesson plans, syllabi, and problem sets. For language models, OERs offer exposure to instructional formatting and domain-specific information, making them valuable for improving performance on knowledge-based downstream tasks. The Common Pile includes a range of such materials sourced from major OER repositories, including collections of open-access books and structured teaching resources.

**Directory of Open Access Books** The Directory of Open Access Books (DOAB) is an online index of over 94,000 peer-reviewed books curated from trusted open-access publishers. To collect the openly licensed content from DOAB, we retrieve metadata using their official metadata feed. We then filter the collection to include only English-language books released under CC BY and CC BY-SA licenses. The filtered books are downloaded in PDF format and converted to plaintext using the Marker PDF-to-text converter. As an additional validation step, we manually create a whitelist of open license statements and retain only texts explicitly containing one of these statements in their front- or back-matter.

**PressBooks** PressBooks is a searchable catalog of over 8,000 open access books. To collect openly licensed content from PressBooks we construct a search query to retrieve URLs for all books written in English and listed as public domain or under CC BY or CC BY-SA licenses. For each matched book, we collect its contents directly from the publicly available web version provided by PressBooks.

**OERCommons** OERCommons is an online platform where educators share open-access instructional materials—such as textbooks, lesson plans, problem sets, course syllabi, and worksheets—with the goal of expanding access to affordable education. To collect the openly licensed content available on OERCommons, we construct a search query to retrieve English-language content released into the public domain or under CC BY or CC BY-SA licenses. The resulting documents are converted to plain text directly from the HTML pages hosted on the OERCommons website.

**LibreTexts** LibreTexts is an online platform that provides a catalog of over 3,000 open-access textbooks. To collect openly licensed content from LibreTexts we gather links to all textbooks in the catalog and check each textbook section for a license statement indicating that it is in the public domain or under a CC BY, CC BY-SA, or the GNU Free Documentation License. We extract plain text from these textbook sections directly from the HTML pages hosted on the LibreTexts website.

## B.7   Wikis

Wikis are collaboratively maintained websites that organize information around specific topics or domains. Their crowd-sourced nature, coupled with community moderation and citation requirements, often results in text that is both informative and well-structured. Prominent examples such as Wikipedia have become staples in large-scale language model pretraining corpora due to their breadth of coverage and high quality. In addition, most major wikis are distributed under open licenses such as CC BY and CC BY-SA. The Common Pile includes content from a range of openly licensed wikis to provide models with structured and well-researched informational text.

**Wikimedia** We downloaded the official database dumps from March 2025 of the English-language wikis that are directly managed by the Wikimedia foundation (see Appendix F for a complete list). These database dumps include the wikitext—Mediawiki's custom markup language—for each page as well as talk pages, where editors discuss changes made to a page. We only use the most recent version of each page. We converted wikitext to plain text using `wtf_wikipedia` after light adjustments in formatting to avoid errors in section ordering caused by a bug. Before parsing, we converted wikitext math into LaTeX math using our custom code. Finally, any remaining HTML tags were removed via regexes.

**Wikiteam** There are many wikis on the internet that are not managed by the Wikimedia foundation, but do use their MediaWiki software to power their wiki. Many of these wikis have been archived by wikiteam, a collection of volunteers that create unofficial database dumps of wikis and upload them to the Internet Archive. We download all dumps made by wikiteam when the metadata indicates the wiki was licensed under CC BY, CC BY-SA, or released into the public domain on the Internet Archive in September of 2024. This results in downloading approximately 330,000 wikis. When multiple dumps of the same wiki exists, we use the most recent dump. The wikitext was converted to plain text following the same steps as with Wikimedia wikis. After preprocessing, we removed documents from wikis that appeared to contain large amounts of license laundering, e.g. those that were collections of song lyrics or transcripts.

## B.8   Source Code

Source code has become an increasingly important component of large-scale language model pretraining corpora, as it enables models to learn syntax, program structure, and problem solving strategies useful for both code generation and reasoning tasks. Thanks to the Free and Open Source Software (FOSS) movement, code also happens to be one of the most openly licensed forms of text, with many software repositories distributed under open licenses such as MIT, BSD, Apache 2.0, and the GNU Free Documentation License (GFDL). The Common Pile includes high-quality, openly licensed source code from large-scale public code datasets and documentation standards, enabling models trained on it to perform better on coding and technical writing tasks.

**The Stack V2** The Stack V2 [114] consists of a mixture of openly licensed and unlicensed work. We use the tooling that the Software Heritage Foundation and BigCode created to build our dataset. In particular, we relied on the license detection performed by the creators of Stack V2. When multiple licenses are detected in a single repository, we make sure that *all* of them meet our definition of "openly licensed".

**Python Enhancement Proposals** Python Enhancement Proposals, or PEPs, are design documents that generally provide a technical specification and rationale for new features of the Python programming language. There are been 661 PEPs published. The majority of PEPs are published in the Public Domain, but 5 were published under the "Open Publication License" and omitted. PEPs are long, highly-polished, and technical in nature and often include code examples paired with their prose. PEPs are authored in ReStructured Text; we used pandoc, version 3.5, to convert them to plain text.

### B.9 Transcribed Audio Content

A historically underutilized source of text data is speech transcribed from audio and video content. Spoken language in educational videos, speeches, and interviews provide an opportunity for models to learn conversational speech patterns.

**Creative Commons YouTube** YouTube is large-scale video-sharing platform where users have the option of uploading content under a CC BY license. To collect high-quality speech-based textual content and combat the rampant license laundering on YouTube, we manually curated a set of over 2,000 YouTube channels that consistently release original openly licensed content containing speech. The resulting collection spans a wide range of genres, including lectures, tutorials, reviews, video essays, speeches, and vlogs. From these channels, we retrieved over 1.1 million openly licensed videos comprising more than 470,000 hours of content. Finally, each video was transcribed to text using the Whisper speech recognition model [139].

### B.10 Web Text

The success of modern LLM pre-training relies on text scraped indiscriminately from the web, as web text covers an extremely diverse range of textual domains. In the Common Pile, we restrict this approach to only include web content with clear public domain status or open license statements.

**Creative Commons Common Crawl** We sourced text from 52 Common Crawl snapshots, covering about half of Common Crawl snapshots available to date and covering all years of operations of Common Crawl up to 2024. We found a higher level of duplication across this collection, suggesting that including more snapshots would lead to a modest increase in total token yield. From these snapshots, we extract HTML content using FastWarc [15]. Then, using a regular expression adapted from the C4Corpus project [69], we retain only those pages where a CC BY, CC BY-SA, or CC0 license appears. To ensure license accuracy, we manually verified the top 1000 domains by content volume, retaining only the 537 domains with confirmed licenses where the Creative Commons designation is applied to all text content rather than only embedded media or a subset of the text on the domain. We extract the main content of these documents and remove boilerplate using Resiliparse [14]. We perform URL-level exact deduplication and use Bloom filters to remove near-duplicates with 80% ngram overlap. We also employ rule-based filters matching Dolma [168]; namely, we use C4-derived heuristics [143] to filter pages containing Javascript, Lorem Ipsum, and curly braces {}. We also apply all Gopher rules [142] to remove low-quality pages. We provide more details on the composition of this subset in Appendix G.

**Foodista** Foodista is a community-maintained site with recipes, food-related news, and nutrition information. All content is licensed under CC BY. Plain text is extracted from the HTML using a custom pipeline that includes extracting title and author information to include at the beginning of the text. Additionally, comments on the page are appended to the article after we filter automatically generated comments.

**News** We scrape the news sites that publish content under CC BY or CC BY-SA according to opennewswire. A full list of sites can be found in Appendix E. Plain text was extracted from the HTML using our custom pipeline, including extraction of the title and byline to include at the beginning of each article.

**Public Domain Review** The Public Domain Review is an online journal dedicated to exploration of works of art and literature that have aged into the public domain. We collect all articles published in the Public Domain Review under a CC BY-SA license.

## C  Additional insights on licensing

There are many standards we could have chosen for what licenses to include in our dataset. The open source, knowledge, and culture movements have harmonized on the high level principles described in section 1: "open" means that permission is granted for content to be freely used, studied, modified, and shared for any purpose. This language is found in the Open Knowledge Definition we follow as well as the Open Source Institute's Open Definition, Creative Commons's statement on Open Culture, Wikimedia's Acceptable licenses policy and more. Our work was also developed to be consistent with the Open movement's work in the specific context of AI technologies such as the

Open Source Initiative's Open Source AI Definition and in consultations with leading members of the community [9].

## C.1 Why we can't always trust automatic license detection

There are many reasons why identifying the licensing status of internet text with automatic tooling can be challenging. In this section, we briefly discuss some major themes from our experience.

**Many AI tools strip out licensing info by accident.**  A serious limitation of using AI tools to determine the licensing of a webpage is that often times the licensing information is located somewhere the models can't see. We found that many web-browsing integrations for LLMs strip out footer and sidebar content that often contains licensing information and that some pretrained models have learned to ignore such text because (for any other task) it's not relevant.

**There are many ways to say the same thing.**  While there exist standards for how to express a license, people don't always follow those standards and failure to follow the standards doesn't mean that the license is invalid. For example, simple string matching on "CC BY" misses a huge amount of CC BY licensed text because a very common way to denote Creative Commons licenses is using an image badge. Current web-processing tools are substantially stronger at identifying text than images, and the failure rate on sites using image badges is quite high.

**Lack of understanding of licenses.**  Most people are not lawyers and do not understand the full legal scope and meaning of the licenses that they attempt to put on their text. Developers routinely tweak boilerplate to produce ambiguous language like ("Licensed under MIT-ish terms") or write contradictory statements ("All rights reserved / CC-BY"). In general, it is common for people to write quasi-legal language along side a more traditional license. Non-standard licenses require substantial amounts of work to interpret and are not always valid or meaningful.

**Licensing signals can be noisy.**  Even when a developer intends to clearly communicate a specific license, contradictions and errors can occur in practice. For example, Longpre et al. [108] found that there were substantial disagreements between the terms of service of a website and the restrictions found in a robots.txt file. We have not yet found a reliable way to have an automatic system identify licensed text and therefore frequently resort to manual review by humans.

## D  List of Data Provenance Initiative sources

The openly licensed supervised datasets included in the Common Pile are listed in Table 1. These datasets were identified and collected using metadata from the Data Provenance Initiative. For more information on these datasets, consult the Data Provenance Initiative Dataset Explorer.

Table 1: Supervised datasets included in the Common Pile from the Data Provenance Initiative collection.

| Collection | Dataset Identifier | Licenses |
|---|---|---|
| AgentInstruct | AgentInstruct-alfworld[165] | MIT License |
| HelpSteer | HelpSteer[185] | CC BY 4.0 |
| Aya Dataset | aya-english[166] | Apache License 2.0 |
| CommitPackFT | commitpackft-abap[166] | MIT License |
| CommitPackFT | commitpackft-agda[166] | MIT License, BSD 3-Clause License |
| CommitPackFT | commitpackft-apl[166] | MIT License, ISC License |
| CommitPackFT | commitpackft-arc[166] | MIT License |
| CommitPackFT | commitpackft-aspectj[166] | Apache License 2.0, BSD 3-Clause License, MIT License |

*Continued on next page*

| Collection | Dataset Identifier | Licenses |
|---|---|---|
| CommitPackFT | commitpackft-ats[166] | Apache License 2.0, MIT License |
| CommitPackFT | commitpackft-blitzmax[166] | MIT License |
| CommitPackFT | commitpackft-bluespec[166] | MIT License |
| CommitPackFT | commitpackft-boo[166] | MIT License |
| CommitPackFT | commitpackft-brainfuck[166] | Apache License 2.0, BSD 2-Clause License, MIT License |
| CommitPackFT | commitpackft-bro[166] | MIT License, BSD 3-Clause License |
| CommitPackFT | commitpackft-cartocss[166] | MIT License |
| CommitPackFT | commitpackft-chapel[166] | Apache License 2.0, BSD 3-Clause License, MIT License |
| CommitPackFT | commitpackft-clean[166] | Apache License 2.0, MIT License |
| CommitPackFT | commitpackft-coldfusion[166] | Apache License 2.0, MIT License |
| CommitPackFT | commitpackft-creole[166] | Apache License 2.0, MIT License |
| CommitPackFT | commitpackft-crystal[166] | Apache License 2.0, MIT License |
| CommitPackFT | commitpackft-dns-zone[166] | MIT License, BSD 3-Clause License |
| CommitPackFT | commitpackft-dylan[166] | MIT License |
| CommitPackFT | commitpackft-eiffel[166] | MIT License |
| CommitPackFT | commitpackft-emberscript[166] | Apache License 2.0, BSD 3-Clause License, MIT License |
| CommitPackFT | commitpackft-fancy[166] | MIT License, BSD 3-Clause License |
| CommitPackFT | commitpackft-flux[166] | Apache License 2.0, MIT License |
| CommitPackFT | commitpackft-forth[166] | MIT License |
| CommitPackFT | commitpackft-g-code[166] | Apache License 2.0, BSD 3-Clause License, MIT License |
| CommitPackFT | commitpackft-gdscript[166] | Apache License 2.0, CC0 1.0, MIT License |
| CommitPackFT | commitpackft-genshi[166] | Apache License 2.0, MIT License |
| CommitPackFT | commitpackft-graphql[166] | Apache License 2.0, BSD 3-Clause License, CC0 1.0, MIT License |
| CommitPackFT | commitpackft-harbour[166] | MIT License |
| CommitPackFT | commitpackft-hlsl[166] | Apache License 2.0, MIT License |
| CommitPackFT | commitpackft-http[166] | Apache License 2.0, MIT License |
| CommitPackFT | commitpackft-idris[166] | MIT License, BSD 3-Clause License, BSD 2-Clause License |
| CommitPackFT | commitpackft-igor-pro[166] | MIT License, BSD 3-Clause License |
| CommitPackFT | commitpackft-inform-7[166] | MIT License, BSD 3-Clause License |
| CommitPackFT | commitpackft-ioke[166] | MIT License |
| CommitPackFT | commitpackft-isabelle[166] | MIT License, BSD 2-Clause License |
| CommitPackFT | commitpackft-jflex[166] | MIT License |
| CommitPackFT | commitpackft-json5[166] | MIT License, BSD 3-Clause License, BSD 2-Clause License |
| CommitPackFT | commitpackft-jsonld[166] | Apache License 2.0, BSD 3-Clause License, CC0 1.0, MIT License |

*Continued on next page*

| Collection | Dataset Identifier | Licenses |
|---|---|---|
| CommitPackFT | commitpackft-krl[166] | MIT License |
| CommitPackFT | commitpackft-latte[166] | MIT License |
| CommitPackFT | commitpackft-lean[166] | Apache License 2.0, MIT License |
| CommitPackFT | commitpackft-lfe[166] | Apache License 2.0, MIT License |
| CommitPackFT | commitpackft-lilypond[166] | MIT License |
| CommitPackFT | commitpackft-liquid[166] | Apache License 2.0, CC0 1.0, MIT License |
| CommitPackFT | commitpackft-literate-agda[166] | MIT License |
| CommitPackFT | commitpackft-literate-coffeescript[166] | MIT License |
| CommitPackFT | commitpackft-literate-haskell[166] | MIT License, BSD 3-Clause License |
| CommitPackFT | commitpackft-llvm[166] | Apache License 2.0, BSD 3-Clause License, BSD 2-Clause License, MIT License |
| CommitPackFT | commitpackft-logos[166] | Apache License 2.0, BSD 3-Clause License, BSD 2-Clause License, MIT License, ISC License |
| CommitPackFT | commitpackft-lsl[166] | MIT License, BSD 3-Clause License |
| CommitPackFT | commitpackft-maple[166] | MIT License, BSD 3-Clause License |
| CommitPackFT | commitpackft-mathematica[166] | MIT License, CC0 1.0 |
| CommitPackFT | commitpackft-metal[166] | Apache License 2.0, MIT License |
| CommitPackFT | commitpackft-mirah[166] | Apache License 2.0, MIT License |
| CommitPackFT | commitpackft-monkey[166] | Apache License 2.0, MIT License |
| CommitPackFT | commitpackft-moonscript[166] | MIT License |
| CommitPackFT | commitpackft-mtml[166] | MIT License |
| CommitPackFT | commitpackft-mupad[166] | Apache License 2.0, BSD 3-Clause License, MIT License |
| CommitPackFT | commitpackft-nesc[166] | MIT License |
| CommitPackFT | commitpackft-netlinx[166] | MIT License |
| CommitPackFT | commitpackft-ninja[166] | Apache License 2.0, BSD 3-Clause License, MIT License |
| CommitPackFT | commitpackft-nit[166] | Apache License 2.0, MIT License |
| CommitPackFT | commitpackft-nu[166] | Apache License 2.0, MIT License |
| CommitPackFT | commitpackft-ooc[166] | MIT License |
| CommitPackFT | commitpackft-openscad[166] | MIT License, CC0 1.0, BSD 2-Clause License |
| CommitPackFT | commitpackft-oz[166] | MIT License, BSD 2-Clause License |
| CommitPackFT | commitpackft-pan[166] | Apache License 2.0, MIT License |
| CommitPackFT | commitpackft-piglatin[166] | Apache License 2.0, MIT License |
| CommitPackFT | commitpackft-pony[166] | MIT License, BSD 2-Clause License |
| CommitPackFT | commitpackft-propeller-spin[166] | MIT License |
| CommitPackFT | commitpackft-pure-data[166] | MIT License |
| CommitPackFT | commitpackft-purebasic[166] | MIT License, BSD 3-Clause License |

*Continued on next page*

| Collection | Dataset Identifier | Licenses |
| --- | --- | --- |
| CommitPackFT | commitpackft-purescript[166] | Apache License 2.0, BSD 3-Clause License, MIT License |
| CommitPackFT | commitpackft-ragel-in-ruby-host[166] | MIT License |
| CommitPackFT | commitpackft-rebol[166] | Apache License 2.0, MIT License |
| CommitPackFT | commitpackft-red[166] | MIT License, BSD 2-Clause License |
| CommitPackFT | commitpackft-rouge[166] | Apache License 2.0, MIT License |
| CommitPackFT | commitpackft-sage[166] | MIT License |
| CommitPackFT | commitpackft-sas[166] | MIT License |
| CommitPackFT | commitpackft-scaml[166] | MIT License, BSD 2-Clause License |
| CommitPackFT | commitpackft-scilab[166] | MIT License, BSD 3-Clause License |
| CommitPackFT | commitpackft-slash[166] | Apache License 2.0, MIT License |
| CommitPackFT | commitpackft-smt[166] | MIT License, BSD 3-Clause License |
| CommitPackFT | commitpackft-solidity[166] | Apache License 2.0, MIT License |
| CommitPackFT | commitpackft-sourcepawn[166] | Apache License 2.0, MIT License |
| CommitPackFT | commitpackft-squirrel[166] | MIT License |
| CommitPackFT | commitpackft-ston[166] | MIT License |
| CommitPackFT | commitpackft-systemverilog[166] | Apache License 2.0, BSD 3-Clause License, MIT License |
| CommitPackFT | commitpackft-unity3d-asset[166] | Apache License 2.0, BSD 3-Clause License, BSD 2-Clause License, MIT License, ISC License, CC0 1.0 |
| CommitPackFT | commitpackft-uno[166] | MIT License |
| CommitPackFT | commitpackft-unrealscript[166] | MIT License |
| CommitPackFT | commitpackft-urweb[166] | MIT License, BSD 3-Clause License |
| CommitPackFT | commitpackft-vcl[166] | Apache License 2.0, BSD 3-Clause License, MIT License |
| CommitPackFT | commitpackft-xbase[166] | Apache License 2.0, MIT License |
| CommitPackFT | commitpackft-xpages[166] | Apache License 2.0, MIT License |
| CommitPackFT | commitpackft-xproc[166] | Apache License 2.0, MIT License |
| CommitPackFT | commitpackft-yacc[166] | MIT License, ISC License, BSD 2-Clause License |
| CommitPackFT | commitpackft-zephir[166] | MIT License |
| CommitPackFT | commitpackft-zig[166] | MIT License |
| Dolly 15k | dolly-brainstorming[166] | CC BY-SA 3.0 |
| Dolly 15k | dolly-classification[166] | CC BY-SA 3.0 |
| Dolly 15k | dolly-closedqa[166] | CC BY-SA 3.0 |
| Dolly 15k | dolly-creative_writing[166] | CC BY-SA 3.0 |
| Dolly 15k | dolly-infoextract[166] | CC BY-SA 3.0 |
| Dolly 15k | dolly-openqa[166] | CC BY-SA 3.0 |
| Dolly 15k | dolly-summarization[166] | CC BY-SA 3.0 |
| DialogStudio | ds-ABCD[166] | Apache License 2.0, MIT License |

*Continued on next page*

| Collection | Dataset Identifier | Licenses |
|---|---|---|
| DialogStudio | ds-ATIS[166] | Apache License 2.0, CC BY 4.0 |
| DialogStudio | ds-ATIS-NER[166] | Apache License 2.0, CC BY 4.0 |
| DialogStudio | ds-AirDialogue[166] | Apache License 2.0 |
| DialogStudio | ds-AntiScam[166] | Apache License 2.0, CC0 1.0 |
| DialogStudio | ds-BANKING77[166] | Apache License 2.0, CC BY 4.0 |
| DialogStudio | ds-BANKING77-OOS[166] | Apache License 2.0, CC BY 4.0 |
| DialogStudio | ds-BiTOD[166] | Apache License 2.0 |
| DialogStudio | ds-CLINC-Single-Domain-OOS-banking[166] | Apache License 2.0, CC BY 3.0 |
| DialogStudio | ds-CLINC-Single-Domain-OOS-credit_cards[166] | Apache License 2.0, CC BY 3.0 |
| DialogStudio | ds-CLINC150[166] | Apache License 2.0, CC BY-SA 3.0 |
| DialogStudio | ds-CaSiNo[166] | Apache License 2.0, CC BY 4.0 |
| DialogStudio | ds-CoQA[166] | Apache License 2.0, MIT License |
| DialogStudio | ds-CoSQL[166] | Apache License 2.0, CC BY-SA 4.0 |
| DialogStudio | ds-ConvAI2[166] | Apache License 2.0 |
| DialogStudio | ds-CraigslistBargains[166] | Apache License 2.0, MIT License |
| DialogStudio | ds-DART[166] | Apache License 2.0, MIT License |
| DialogStudio | ds-DSTC8-SGD[166] | Apache License 2.0, CC BY-SA 4.0 |
| DialogStudio | ds-DialogSum[166] | Apache License 2.0, MIT License |
| DialogStudio | ds-Disambiguation[166] | Apache License 2.0, MIT License |
| DialogStudio | ds-FeTaQA[166] | Apache License 2.0, CC BY-SA 4.0 |
| DialogStudio | ds-GECOR[166] | Apache License 2.0, CC BY 4.0 |
| DialogStudio | ds-GrailQA[166] | Apache License 2.0 |
| DialogStudio | ds-HDSA-Dialog[166] | Apache License 2.0, MIT License |
| DialogStudio | ds-HH-RLHF[166] | Apache License 2.0, MIT License |
| DialogStudio | ds-HWU64[166] | Apache License 2.0, CC BY-SA 3.0 |
| DialogStudio | ds-HybridQA[166] | Apache License 2.0, MIT License |
| DialogStudio | ds-KETOD[166] | Apache License 2.0, MIT License |
| DialogStudio | ds-MTOP[166] | Apache License 2.0, CC BY-SA 4.0 |
| DialogStudio | ds-MULTIWOZ2_2[166] | Apache License 2.0, MIT License |
| DialogStudio | ds-MulDoGO[166] | Apache License 2.0, CDLA Permissive 1.0 |
| DialogStudio | ds-MultiWOZ_2.1[166] | Apache License 2.0, MIT License |
| DialogStudio | ds-Prosocial[166] | Apache License 2.0, MIT License |
| DialogStudio | ds-RESTAURANTS8K[166] | Apache License 2.0, CC BY 4.0 |
| DialogStudio | ds-SGD[166] | Apache License 2.0, CC BY-SA 4.0 |
| DialogStudio | ds-SNIPS[166] | Apache License 2.0 |
| DialogStudio | ds-SNIPS-NER[166] | Apache License 2.0 |
| DialogStudio | ds-SParC[166] | Apache License 2.0, CC BY-SA 4.0 |
| DialogStudio | ds-SQA[166] | Apache License 2.0, CC BY-SA 4.0 |

*Continued on next page*

| Collection | Dataset Identifier | Licenses |
|---|---|---|
| DialogStudio | ds-STAR[166] | Apache License 2.0, MIT License |
| DialogStudio | ds-Spider[166] | Apache License 2.0, CC BY-SA 4.0 |
| DialogStudio | ds-TOP[166] | Apache License 2.0, CC BY-SA |
| DialogStudio | ds-TOP-NER[166] | Apache License 2.0, CC BY-SA |
| DialogStudio | ds-Taskmaster1[166] | Apache License 2.0, CC BY 4.0 |
| DialogStudio | ds-Taskmaster2[166] | Apache License 2.0, CC BY 4.0 |
| DialogStudio | ds-Taskmaster3[166] | Apache License 2.0, CC BY 4.0 |
| DialogStudio | ds-ToTTo[166] | Apache License 2.0, CC BY-SA 3.0 |
| DialogStudio | ds-TweetSumm[166] | Apache License 2.0, CC0 1.0 |
| DialogStudio | ds-WOZ2_0[166] | Apache License 2.0 |
| DialogStudio | ds-WebQSP[166] | Apache License 2.0, CC BY 4.0 |
| DialogStudio | ds-WikiSQL[166] | Apache License 2.0, BSD 3-Clause License |
| DialogStudio | ds-WikiTQ[166] | Apache License 2.0, CC BY-SA 4.0 |
| DialogStudio | ds-chitchat-dataset[166] | Apache License 2.0, MIT License |
| DialogStudio | ds-wizard_of_internet[166] | Apache License 2.0, CC BY 4.0 |
| DialogStudio | ds-wizard_of_wikipedia[166] | Apache License 2.0, CC BY 4.0 |
| Flan Collection (Chain-of-Thought) | fc-cot-cot_gsm8k[34] | MIT License |
| Flan Collection (Chain-of-Thought) | fc-cot-cot_strategyqa[61] | CC BY-SA 3.0 |
| Flan Collection (Chain-of-Thought) | fc-cot-stream_creak[126] | MIT License, CC BY-SA 4.0 |
| Flan Collection (Chain-of-Thought) | fc-cot-stream_esnli[21] | MIT License, CC BY-SA 4.0 |
| Flan Collection (Flan 2021) | fc-flan-drop[47] | CC BY 4.0 |
| Flan Collection (Flan 2021) | fc-flan-e2e_nlg[124] | CC BY-SA 4.0 |
| Flan Collection (Flan 2021) | fc-flan-natural_questions[90] | Apache License 2.0, CC BY-SA 3.0 |
| Flan Collection (Flan 2021) | fc-flan-quac[29] | CC BY-SA 4.0 |
| Flan Collection (Flan 2021) | fc-flan-squad_v1[148] | CC BY-SA 4.0 |
| Flan Collection (Flan 2021) | fc-flan-squad_v2[148] | CC BY-SA 4.0 |
| Flan Collection (Flan 2021) | fc-flan-trec[101] | CC0 1.0 |
| Flan Collection (Flan 2021) | fc-flan-true_case[101] | CC0 1.0 |
| Flan Collection (Flan 2021) | fc-flan-wiki_lingua_english_en[91] | CC BY 3.0 |
| Flan Collection (Flan 2021) | fc-flan-winogrande[159] | Apache License 2.0, CC BY 4.0 |
| Flan Collection (Flan 2021) | fc-flan-wnli[99] | CC BY 4.0 |
| Flan Collection (Flan 2021) | fc-flan-word_segment[99] | CC0 1.0 |
| Flan Collection (Flan 2021) | fc-flan-wsc[99] | CC BY 4.0 |
| Flan Collection (P3) | fc-p3-adversarial_qa[11] | CC BY-SA 3.0 |
| Flan Collection (P3) | fc-p3-cos_e[145] | BSD 3-Clause License |
| Flan Collection (P3) | fc-p3-dbpedia_14[98] | CC BY-SA 3.0 |
| Flan Collection (P3) | fc-p3-hotpotqa[191] | Apache License 2.0, CC BY-SA 4.0 |
| Flan Collection (P3) | fc-p3-quarel[154] | CC BY 4.0 |
| Flan Collection (P3) | fc-p3-quartz[171] | CC BY 4.0 |

*Continued on next page*

| Collection | Dataset Identifier | Licenses |
|---|---|---|
| Flan Collection (P3) | fc-p3-quoref[45] | CC BY 4.0 |
| Flan Collection (P3) | fc-p3-web_questions[13] | CC BY 4.0 |
| Flan Collection (P3) | fc-p3-wiki_bio[94] | CC BY-SA 3.0 |
| Flan Collection (P3) | fc-p3-wiki_hop[94] | CC BY-SA 3.0 |
| Flan Collection (Super-NaturalInstructions) | fc-sni-adversarial_qa[11] | CC BY-SA 3.0 |
| Flan Collection (Super-NaturalInstructions) | fc-sni-adverserial_qa[11] | MIT License |
| Flan Collection (Super-NaturalInstructions) | fc-sni-air_dialogue[188] | Apache License 2.0 |
| Flan Collection (Super-NaturalInstructions) | fc-sni-ancora_ca_ner[188] | CC BY 4.0 |
| Flan Collection (Super-NaturalInstructions) | fc-sni-anem[125] | MIT License, CC BY-SA 3.0 |
| Flan Collection (Super-NaturalInstructions) | fc-sni-argkp | Apache License 2.0, CC BY-SA 3.0 |
| Flan Collection (Super-NaturalInstructions) | fc-sni-asian_language_-treebank[152] | CC BY 4.0 |
| Flan Collection (Super-NaturalInstructions) | fc-sni-atomic[76] | CC BY 4.0 |
| Flan Collection (Super-NaturalInstructions) | fc-sni-bard[55] | Apache License 2.0 |
| Flan Collection (Super-NaturalInstructions) | fc-sni-cedr[162] | Apache License 2.0 |
| Flan Collection (Super-NaturalInstructions) | fc-sni-circa[113] | CC BY-SA 4.0 |
| Flan Collection (Super-NaturalInstructions) | fc-sni-clue_cmrc2018[43] | CC BY-SA 4.0 |
| Flan Collection (Super-NaturalInstructions) | fc-sni-coached_conv_pref[140] | CC BY 4.0 |
| Flan Collection (Super-NaturalInstructions) | fc-sni-copa_hr | BSD 2-Clause License |
| Flan Collection (Super-NaturalInstructions) | fc-sni-crows_pairs[123] | CC BY-SA 4.0 |
| Flan Collection (Super-NaturalInstructions) | fc-sni-cuad[72] | CC BY 4.0 |
| Flan Collection (Super-NaturalInstructions) | fc-sni-defeasible_nli_atomic[156] | MIT License |
| Flan Collection (Super-NaturalInstructions) | fc-sni-disfl_qa[68] | CC BY 4.0 |
| Flan Collection (Super-NaturalInstructions) | fc-sni-e_snli[21] | MIT License |
| Flan Collection (Super-NaturalInstructions) | fc-sni-gap[187] | Apache License 2.0 |
| Flan Collection (Super-NaturalInstructions) | fc-sni-hotpotqa[191] | Apache License 2.0, CC BY-SA 4.0 |

*Continued on next page*

| Collection | Dataset Identifier | Licenses |
|---|---|---|
| Flan Collection (Super-NaturalInstructions) | fc-sni-human_ratings_of_natural_-language_generation_outputs[191] | CC BY 4.0 |
| Flan Collection (Super-NaturalInstructions) | fc-sni-hybridqa[26] | CC BY 4.0, MIT License |
| Flan Collection (Super-NaturalInstructions) | fc-sni-iirc[54] | CC BY 4.0 |
| Flan Collection (Super-NaturalInstructions) | fc-sni-jigsaw[54] | CC0 1.0 |
| Flan Collection (Super-NaturalInstructions) | fc-sni-librispeech_asr[128] | CC BY 4.0 |
| Flan Collection (Super-NaturalInstructions) | fc-sni-logic2text[27] | MIT License |
| Flan Collection (Super-NaturalInstructions) | fc-sni-numeric_fused_head[50] | MIT License |
| Flan Collection (Super-NaturalInstructions) | fc-sni-offenseval_dravidian[50] | CC BY 4.0 |
| Flan Collection (Super-NaturalInstructions) | fc-sni-open_pi[173] | CC BY 4.0 |
| Flan Collection (Super-NaturalInstructions) | fc-sni-paper_reviews_data_set[173] | CC BY 4.0 |
| Flan Collection (Super-NaturalInstructions) | fc-sni-poem_sentiment[164] | CC BY 4.0 |
| Flan Collection (Super-NaturalInstructions) | fc-sni-propara[44] | Apache License 2.0 |
| Flan Collection (Super-NaturalInstructions) | fc-sni-quarel[154] | CC BY 4.0 |
| Flan Collection (Super-NaturalInstructions) | fc-sni-quartz[171] | CC BY 4.0 |
| Flan Collection (Super-NaturalInstructions) | fc-sni-quoref[45] | CC BY 4.0 |
| Flan Collection (Super-NaturalInstructions) | fc-sni-ro_sts_parallel[49] | CC BY-SA 4.0 |
| Flan Collection (Super-NaturalInstructions) | fc-sni-schema_guided_dstc8[149] | CC BY-SA 4.0 |
| Flan Collection (Super-NaturalInstructions) | fc-sni-scitail[87] | Apache License 2.0 |
| Flan Collection (Super-NaturalInstructions) | fc-sni-scitailv1.1[87] | Apache License 2.0 |
| Flan Collection (Super-NaturalInstructions) | fc-sni-semeval_2020_task4[87] | CC BY-SA 4.0 |
| Flan Collection (Super-NaturalInstructions) | fc-sni-sms_spam_collection_v.1[87] | CC BY 4.0 |
| Flan Collection (Super-NaturalInstructions) | fc-sni-splash[87] | CC BY-SA 4.0 |
| Flan Collection (Super-NaturalInstructions) | fc-sni-squad2.0[148] | CC BY-SA 4.0 |
| Flan Collection (Super-NaturalInstructions) | fc-sni-squad_1.1[147] | CC BY-SA 4.0 |

| Collection | Dataset Identifier | Licenses |
|---|---|---|
| Flan Collection (Super-NaturalInstructions) | fc-sni-strategyqa[61] | MIT License |
| Flan Collection (Super-NaturalInstructions) | fc-sni-universal_dependencies___-english_dependency_treebank[61] | CC BY-SA 4.0 |
| Flan Collection (Super-NaturalInstructions) | fc-sni-web_questions[13] | CC BY 4.0 |
| Flan Collection (Super-NaturalInstructions) | fc-sni-wiki_hop[94] | CC BY-SA 3.0 |
| Flan Collection (Super-NaturalInstructions) | fc-sni-wikitext[117] | CC BY-SA 3.0 |
| Flan Collection (Super-NaturalInstructions) | fc-sni-winograd_wsc[99] | CC BY 4.0 |
| Flan Collection (Super-NaturalInstructions) | fc-sni-winomt[169] | MIT License |
| Flan Collection (Super-NaturalInstructions) | fc-sni-winowhy[195] | MIT License |
| Flan Collection (Super-NaturalInstructions) | fc-sni-wsc; enhanced_wsc[195] | CC BY 4.0 |
| Flan Collection (Super-NaturalInstructions) | fc-sni-wsc_fiexed[195] | CC BY-SA 3.0 |
| Flan Collection (Super-NaturalInstructions) | fc-sni-xcopa[135] | CC BY 4.0 |
| Flan Collection (Super-NaturalInstructions) | fc-sni-xquad[6] | CC BY-SA 4.0 |
| Open Assistant | oasst-en[89] | Apache License 2.0, CC BY 4.0 |
| Open Assistant OctoPack | oasst-en-octopack[89] | Apache License 2.0, CC BY 4.0 |
| Open Assistant v2 | oasst2-en[89] | Apache License 2.0 |
| OIG | oig-unified_canadian_-parliament[89] | Apache License 2.0 |
| OIG | oig-unified_cuad[89] | Apache License 2.0, CC BY 4.0 |
| OIG | oig-unified_grade_school_math_-instructions[89] | Apache License 2.0, MIT License |
| OIG | oig-unified_nq[89] | Apache License 2.0, CC BY-SA 3.0 |
| OIG | oig-unified_sqlv1[89] | Apache License 2.0, CC BY-SA 4.0 |
| OIG | oig-unified_sqlv2[89] | Apache License 2.0, CC BY-SA 4.0 |
| OIG | oig-unified_squad_v2_more_-neg[89] | Apache License 2.0, CC BY-SA 4.0 |
| Tasksource Instruct | tsi-balanced_copa[86] | BSD 2-Clause License |
| Tasksource Instruct | tsi-breaking_nli[62] | CC BY-SA 4.0 |
| Tasksource Instruct | tsi-cladder[62] | MIT License |
| Tasksource Instruct | tsi-condaqa[150] | Apache License 2.0 |
| Tasksource Instruct | tsi-conj_nli[158] | MIT License |
| Tasksource Instruct | tsi-defeasible_nli-atomic[156] | MIT License |
| Tasksource Instruct | tsi-defeasible_nli-snli[156] | MIT License |

*Continued on next page*

| Collection | Dataset Identifier | Licenses |
|---|---|---|
| Tasksource Instruct | tsi-dynasent-dynabench.dynasent.r1.all-r1[136] | CC BY 4.0 |
| Tasksource Instruct | tsi-dynasent-dynabench.dynasent.r2.all-r2[136] | CC BY 4.0 |
| Tasksource Instruct | tsi-fever_evidence_related-mwong_-_fever_related[178] | CC BY-SA 4.0 |
| Tasksource Instruct | tsi-few_nerd-supervised[46] | CC BY-SA 4.0 |
| Tasksource Instruct | tsi-fig_qa[103] | MIT License |
| Tasksource Instruct | tsi-fracas[57] | MIT License |
| Tasksource Instruct | tsi-hyperpartisan_news[57] | CC BY 4.0 |
| Tasksource Instruct | tsi-lex_glue-case_hold[23] | Apache License 2.0 |
| Tasksource Instruct | tsi-lonli[175] | MIT License |
| Tasksource Instruct | tsi-moral_stories-full[52] | MIT License |
| Tasksource Instruct | tsi-neqa[52] | CC BY 4.0 |
| Tasksource Instruct | tsi-prost | Apache License 2.0 |
| Tasksource Instruct | tsi-quote_repetition | CC BY 4.0 |
| Tasksource Instruct | tsi-recast-recast_factuality | CC BY-SA 4.0 |
| Tasksource Instruct | tsi-recast-recast_megaveridicality | CC BY-SA 4.0 |
| Tasksource Instruct | tsi-recast-recast_ner | CC BY-SA 4.0 |
| Tasksource Instruct | tsi-recast-recast_puns | CC BY-SA 4.0 |
| Tasksource Instruct | tsi-recast-recast_sentiment | CC BY-SA 4.0 |
| Tasksource Instruct | tsi-recast-recast_verbcorner | CC BY-SA 4.0 |
| Tasksource Instruct | tsi-recast-recast_verbnet | CC BY-SA 4.0 |
| Tasksource Instruct | tsi-redefine_math | CC BY 4.0 |
| Tasksource Instruct | tsi-tracie[197] | Apache License 2.0 |
| Tasksource Instruct | tsi-truthful_qa-multiple_-choice[102] | Apache License 2.0 |
| Tasksource Instruct | tsi-vitaminc-tals__vitaminc[163] | MIT License |
| Tasksource Instruct | tsi-winowhy[195] | MIT License |
| Tasksource Symbol-Tuning | tsy-breaking_nli[62] | CC BY-SA 4.0 |
| Tasksource Symbol-Tuning | tsy-cladder[62] | MIT License |
| Tasksource Symbol-Tuning | tsy-condaqa[150] | Apache License 2.0 |
| Tasksource Symbol-Tuning | tsy-conj_nli[158] | MIT License |
| Tasksource Symbol-Tuning | tsy-defeasible_nli-atomic[156] | MIT License |
| Tasksource Symbol-Tuning | tsy-defeasible_nli-snli[156] | MIT License |
| Tasksource Symbol-Tuning | tsy-dynasent-dynabench.dynasent.r1.all-r1[136] | CC BY 4.0 |
| Tasksource Symbol-Tuning | tsy-dynasent-dynabench.dynasent.r2.all-r2[136] | CC BY 4.0 |
| Tasksource Symbol-Tuning | tsy-fever_evidence_related-mwong__fever_related[178] | CC BY-SA 4.0 |
| Tasksource Symbol-Tuning | tsy-fracas[57] | MIT License |

*Continued on next page*

| Collection | Dataset Identifier | Licenses |
|---|---|---|
| Tasksource Symbol-Tuning | tsy-hyperpartisan_news[57] | CC BY 4.0 |
| Tasksource Symbol-Tuning | tsy-lonli[175] | MIT License |
| Tasksource Symbol-Tuning | tsy-recast-recast_factuality[175] | CC BY-SA 4.0 |
| Tasksource Symbol-Tuning | tsy-recast-recast_-megaveridicality[175] | CC BY-SA 4.0 |
| Tasksource Symbol-Tuning | tsy-recast-recast_ner[175] | CC BY-SA 4.0 |
| Tasksource Symbol-Tuning | tsy-recast-recast_puns[175] | CC BY-SA 4.0 |
| Tasksource Symbol-Tuning | tsy-recast-recast_sentiment[175] | CC BY-SA 4.0 |
| Tasksource Symbol-Tuning | tsy-recast-recast_verbcorner[175] | CC BY-SA 4.0 |
| Tasksource Symbol-Tuning | tsy-recast-recast_verbnet[175] | CC BY-SA 4.0 |
| Tasksource Symbol-Tuning | tsy-tracie[197] | Apache License 2.0 |
| Tasksource Symbol-Tuning | tsy-vitaminc-tals__vitaminc[163] | MIT License |
| Tasksource Symbol-Tuning | tsy-winowhy[195] | MIT License |

# E  List of News sources

The Common Pile contains a variety of openly licensed news sources released under CC BY and CC BY-SA licenses. The sources licensed under CC BY include: 360info, Africa is a Country, Alt News, Balkan Diskurs, Factly, Freedom of the Press Foundation, Agenzia Fides, Global Voices, Meduza, Mekong Eye, Milwaukee Neighborhood News Service, Minority Africa, New Canadian Media, SciDev.Net, The Solutions Journalism Exchange, Tasnim News Agency, and ZimFact. The sources licensed under CC BY-SA include: Oxpeckers, Propastop, and The Public Record.

# F  List of WikiMedia wikis

Official Wikimedia wikis are released under a CC BY-SA license. The Common Pile includes the following Wikimedia wikis: Wikipedia, Wikinews, Wikibooks, Wikiquote, Wikisource, Wikiversity, Wikivoyage, and Wiktionary.

# G  CCCC Source Statistics

We provide additional statistics on the CCCC subset of the Common Pile, including the number of unicode words and documents sourced from each Common Crawl snapshot, in Table 2.

Table 2: Counts of words and documents extracted from 52 snapshots after filtering with our pipeline.

| Snapshot | Unicode Words | Documents |
|---|---|---|
| CC-MAIN-2013-20 | 3,851,018,197 | 5,529,294 |
| CC-MAIN-2013-48 | 4,544,197,252 | 6,997,831 |
| CC-MAIN-2014-10 | 4,429,217,941 | 6,682,672 |
| CC-MAIN-2014-15 | 4,059,132,873 | 5,912,779 |
| CC-MAIN-2014-23 | 5,193,195,765 | 8,253,690 |
| *Continued on next page* | | |

| Snapshot | Unicode Words | Documents |
|---|---|---|
| CC-MAIN-2014-35 | 4,254,690,945 | 6,551,673 |
| CC-MAIN-2014-41 | 4,289,814,449 | 6,558,170 |
| CC-MAIN-2014-42 | 3,986,284,741 | 6,144,797 |
| CC-MAIN-2014-49 | 3,316,075,452 | 4,699,472 |
| CC-MAIN-2014-52 | 4,307,765,289 | 6,338,983 |
| CC-MAIN-2015-06 | 3,675,982,679 | 5,181,955 |
| CC-MAIN-2015-11 | 3,932,442,900 | 5,438,533 |
| CC-MAIN-2015-14 | 3,658,107,765 | 4,954,273 |
| CC-MAIN-2015-18 | 4,451,734,946 | 6,319,757 |
| CC-MAIN-2015-22 | 4,285,945,319 | 5,949,267 |
| CC-MAIN-2015-27 | 3,639,904,128 | 4,975,152 |
| CC-MAIN-2016-07 | 1,588,496,703 | 3,798,207 |
| CC-MAIN-2016-18 | 3,228,754,200 | 4,446,815 |
| CC-MAIN-2016-22 | 3,217,827,676 | 4,242,762 |
| CC-MAIN-2017-04 | 3,852,699,213 | 5,239,605 |
| CC-MAIN-2017-09 | 4,186,915,498 | 5,119,171 |
| CC-MAIN-2017-13 | 4,950,110,931 | 5,923,670 |
| CC-MAIN-2017-17 | 4,684,050,830 | 5,645,725 |
| CC-MAIN-2017-22 | 4,683,569,278 | 5,514,717 |
| CC-MAIN-2017-26 | 4,744,689,137 | 5,514,047 |
| CC-MAIN-2017-51 | 1,981,004,306 | 2,529,289 |
| CC-MAIN-2018-13 | 4,816,417,930 | 5,520,099 |
| CC-MAIN-2018-22 | 3,921,533,251 | 4,401,956 |
| CC-MAIN-2018-26 | 4,506,583,931 | 4,916,546 |
| CC-MAIN-2018-30 | 4,936,722,403 | 5,282,886 |
| CC-MAIN-2018-34 | 3,865,953,978 | 3,808,725 |
| CC-MAIN-2018-47 | 3,933,439,841 | 3,637,947 |
| CC-MAIN-2018-51 | 4,745,124,422 | 4,616,832 |
| CC-MAIN-2019-04 | 4,475,679,190 | 4,140,277 |
| CC-MAIN-2019-09 | 4,287,868,800 | 4,142,190 |
| CC-MAIN-2019-13 | 3,966,330,348 | 3,849,631 |
| CC-MAIN-2019-30 | 4,179,526,188 | 4,430,572 |
| CC-MAIN-2019-35 | 5,144,426,270 | 5,048,106 |
| CC-MAIN-2019-39 | 4,572,972,457 | 4,527,430 |
| CC-MAIN-2020-29 | 5,200,565,501 | 4,984,248 |
| CC-MAIN-2020-34 | 4,458,827,947 | 4,297,009 |
| CC-MAIN-2021-17 | 1,768,757,386 | 1,824,942 |
| CC-MAIN-2021-39 | 4,599,961,675 | 4,287,356 |
| CC-MAIN-2021-43 | 5,337,349,331 | 5,304,846 |
| CC-MAIN-2021-49 | 3,980,018,773 | 4,050,641 |

| Snapshot | Unicode Words | Documents |
|---|---|---|
| CC-MAIN-2022-05 | 4,517,850,019 | 4,503,863 |
| CC-MAIN-2023-06 | 5,135,614,227 | 4,959,915 |
| CC-MAIN-2023-14 | 5,117,143,765 | 4,675,097 |
| CC-MAIN-2023-23 | 5,461,486,807 | 4,869,627 |
| CC-MAIN-2023-50 | 5,881,860,014 | 4,901,306 |
| CC-MAIN-2024-10 | 5,164,171,562 | 4,335,071 |
| CC-MAIN-2024-18 | 4,745,457,054 | 3,949,186 |
| **Total** | **221,715,271,483** | **259,728,610** |

## H  PeS2o Source Statistics

Additional statistics on the composition of the peS2o subset of the Common Pile can be found in Table 3 and Table 4.

Table 3: Distribution of licenses in the peS2o subset.

| License | Train Split | Validation Split |
|---|---|---|
| CC BY | 6,088,325 | 37,754 |
| CC BY-SA | 120,150 | 1,231 |
| CC0 | 36,373 | 121 |
| Public domain | 10,060 | 6 |

Table 4: Distribution of papers across 23 fields of study, as identified by the Semantic Scholar API [88]. A paper may belong to one or more fields of study.

| Field of Study | Train Split | Validation Split |
|---|---|---|
| Medicine | 2,435,244 | 23,734 |
| Biology | 1,518,478 | 8,879 |
| Environmental Science | 993,499 | 7,601 |
| Engineering | 656,021 | 5,005 |
| Computer Science | 462,320 | 3,003 |
| Materials Science | 416,045 | 3,166 |
| Physics | 413,461 | 1,285 |
| Chemistry | 406,429 | 2,781 |
| Psychology | 364,441 | 2,126 |
| Education | 220,014 | 1,532 |
| Business | 193,536 | 946 |
| Economics | 185,716 | 921 |

*Continued on next page*

| Field of Study | Train Split | Validation Split |
|---|---|---|
| Agricultural and Food Sciences | 333,776 | 2,013 |
| Sociology | 137,257 | 1,535 |
| Mathematics | 135,676 | 199 |
| Political Science | 106,748 | 378 |
| Geology | 67,258 | 217 |
| Geography | 44,269 | 257 |
| Linguistics | 41,737 | 228 |
| History | 36,848 | 192 |
| Law | 30,888 | 251 |
| Philosophy | 27,518 | 148 |
| Art | 26,658 | 75 |

# I   Growth rates of openly licensed data

Over time, the volume of openly licensed data continues to grow as more creators release content under open licenses. In Figure 6, we quantify this growth between 2010 and 2024 by analyzing subsets of the Common Pile for which reliable creation date metadata is available. We plot the cumulative proportion of data created up to various cutoff dates and find that approximately half of the Common Pile (around 3.8TB) was created since 2020. This trend provides insight into the growing availability of openly licensed data and suggests a promising trajectory for future LLMs trained entirely on openly licensed sources.

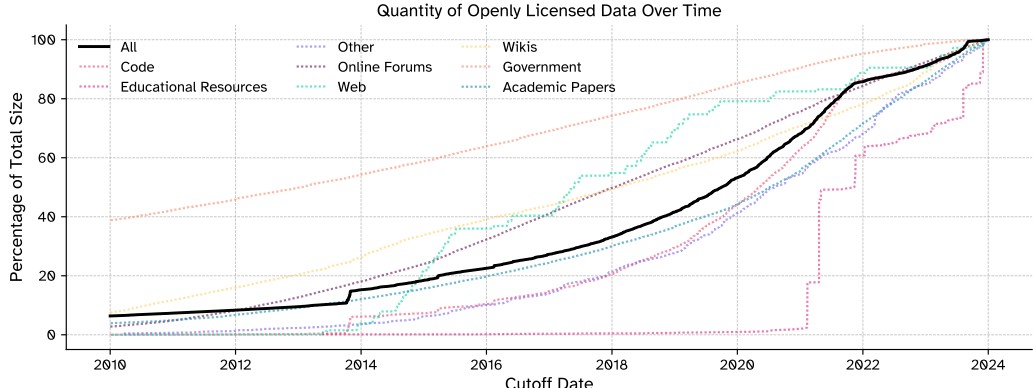

Figure 6: **The amount of openly licensed text grows steadily over time.** We visualize the cumulative proportion of data created up to various cutoff dates for sources in the Common Pile with reliable creation date metadata. This includes all sources except for the Caselaw Access Project, Data Provenance Initiative, and the sources covering early 20th century Public Domain books.

# J   Details on filtering pipelines

In subsection 4.1, we detail the steps used to produce the Comma v0.1 training dataset from the raw text in the Common Pile. These include applying filters based on language, text quality, length, likelihood, and toxicity; removing various forms of PII; and removal of source-specific boilerplate

text using regular expressions. The Common Pile contains a diverse range of sources and we therefore design separate filtering thresholds for each source. The exact source-specific thresholds used to post-process the Common Pile can be found in Table 5. Additionally, statistics on the pre- and post-filtered sizes of each source can be found in Table 6.

Table 5: Pre-processing pipelines applied to each source in the Common Pile to construct the Comma dataset.

| Source | Language | Text Quality | Doc Length | Log-Likelihood | Toxicity | PII | Regex Filter |
|---|---|---|---|---|---|---|---|
| ArXiv Abstracts | – | – | – | – | – | Y | N |
| ArXiv Papers | > 0.5 | – | – | – | – | Y | N |
| Biodiversity Heritage Library | > 0.5 | – | > 100 | > -20 | – | N | Y |
| Caselaw Access Project | – | – | > 100 | – | > 0.1 | Y | N |
| CC Common Crawl | > 0.5 | > 0.0001 | > 100 | – | > 0.1 | Y | N |
| Data Provenance Initiative | – | – | – | – | – | N | N |
| Database of Open Access Books | > 0.5 | – | > 200 | – | > 0.1 | Y | N |
| Foodista | > 0.5 | – | > 100 | – | – | N | N |
| GitHub Archive | > 0.5 | – | > 100 | – | > 0.1 | Y | N |
| Library of Congress | – | – | – | > -20 | > 0.1 | N | Y |
| LibreTexts | > 0.5 | – | > 700 | – | > 0.1 | Y | N |
| News | > 0.5 | – | > 100 | – | – | Y | N |
| OERCommons | > 0.5 | – | > 300 | – | > 0.1 | Y | N |
| peS2o | – | – | – | – | – | Y | N |
| Pre-1929 Books | – | – | – | > -20 | > 0.1 | N | Y |
| PressBooks | > 0.5 | – | > 600 | – | > 0.1 | Y | N |
| Project Gutenberg | > 0.5 | – | – | > -20 | – | N | N |
| Public Domain Review | – | – | > 100 | – | – | Y | N |
| PubMed | > 0.5 | – | > 100 | – | – | Y | N |
| PEPs | – | – | – | – | – | Y | N |
| Regulations.gov | – | – | > 100 | – | – | Y | Y |
| StackExchange | > 0.5 | – | – | – | – | Y | N |
| Ubuntu IRC | > 0.5 | – | > 100 | – | > 0.1 | Y | N |
| UK Hansard | > 0.5 | – | – | – | – | Y | N |
| USGPO | – | – | – | – | – | N | Y |
| USPTO | – | – | > 100 | > -20 | – | Y | N |
| Wikimedia | > 0.5 | – | > 100 | – | – | Y | N |
| Wikiteam | > 0.5 | – | > 700 | – | > 0.1 | Y | N |

*Continued on next page*

| Source | Language | Text Quality | Doc Length | Log-Likelihood | Toxicity | PII | Regex Filter |
|--------|----------|--------------|------------|----------------|----------|-----|--------------|
| CC YouTube | > 0.5 | – | > 100 | – | > 0.1 | Y | N |

Table 6: Raw and filtered sizes of the Common Pile's constituent datasets.

| | Document Count | | Size (GB) | |
|--------|-----|----------|-----|----------|
| Source | Raw | Filtered | Raw | Filtered |
| ArXiv Abstracts | 2,538,935 | 2,504,679 | 2.4 | 2.4 |
| ArXiv Papers | 321,336 | 304,048 | 21 | 19 |
| Biodiversity Heritage Library | 42,418,498 | 15,111,313 | 96 | 35 |
| Caselaw Access Project | 6,919,240 | 6,735,525 | 78 | 77 |
| CC Common Crawl | 51,054,412 | 6,852,137 | 260 | 58 |
| Data Provenance Initiative | 9,688,211 | 3,508,518 | 7 | 3 |
| Directory of Open Access Books | 474,445 | 403,992 | 12.5 | 12 |
| Foodista | 72,090 | 65,640 | 0.09 | 0.08 |
| GitHub Archive | 30,318,774 | 23,358,580 | 54.7 | 40.4 |
| Library of Congress | 135,500 | 129,052 | 47.8 | 35.6 |
| LibreTexts | 62,269 | 40,049 | 5.3 | 3.6 |
| News | 172,308 | 126,673 | 0.4 | 0.3 |
| OERCommons | 9,339 | 5,249 | 0.1 | 0.05 |
| peS2o | 6,294,020 | 6,117,280 | 188.2 | 182.6 |
| Pre-1929 Books | 137,127 | 124,898 | 73.8 | 46.3 |
| PressBooks | 106,881 | 54,455 | 1.5 | 0.6 |
| Project Gutenberg | 71,810 | 55,454 | 26.2 | 20.1 |
| Public Domain Review | 1,412 | 1,406 | 0.007 | 0.007 |
| PubMed | 4,068,867 | 3,829,689 | 158.9 | 147.1 |
| PEPs | 656 | 655 | 0.01 | 0.01 |
| Regulations.gov | 225,196 | 208,301 | 6.1 | 5.1 |
| StackExchange | 33,415,400 | 30,987,814 | 103.7 | 89.7 |
| Stack V2 | 218,364,133 | 69,588,607 | 4774.7 | 259.9 |
| Ubuntu IRC | 329,115 | 234,982 | 6.3 | 5.3 |
| UK Hansard | 51,552 | 47,909 | 10 | 9.6 |
| USGPO | 2,732,677 | 2,148,548 | 74.5 | 36.1 |
| USPTO | 20,294,152 | 17,030,231 | 1003.4 | 661.1 |
| Wikimedia | 63,969,938 | 16,311,574 | 90.5 | 57.4 |
| Wikiteam | 219,139,368 | 26,931,807 | 437.5 | 13.7 |
| CC YouTube | 1,129,692 | 998,104 | 21.5 | 18.6 |

*Continued on next page*

| Source | Document Count | | Size (GB) | |
|---|---|---|---|---|
| | Raw | Filtered | Raw | Filtered |
| **Total** | **692,854,953** | **233,817,169** | **7557.9** | **1838.3** |

# K   Details on Comma's pre-training data mixture

We estimated the quality of each source in the Common Pile by training a 1.7B-parameter model for 28B tokens on each source individually and evaluating the resulting models on the set of "early signal" tasks from [133]. In doing so, we found that the amount of text in each source was poorly correlated with text quality, motivating the use of heuristic mixing weights to up-/down-weight different sources in our pre-training mix. In Table 7 we list the pre-training mixture weights for each of the sources in the Common Pile.

Table 7: Overview of the data mixing used to up/down-weight individual sources in the Common Pile to construct the Comma v0.1-1T pre-training dataset. Comma v0.1-2T simply repeats this full mixture twice.

| Source | Size (GB) | Repeats | Effective Size (GB) | Tokens (Billions) | Percentage |
|---|---|---|---|---|---|
| ArXiv Abstracts | 2.4 | 6 | 14.4 | 3.6 | 0.360% |
| ArXiv Papers | 19.5 | 6 | 117 | 29.3 | 2.932% |
| Biodiversity Heritage Library | 35.5 | 0.25 | 8.9 | 2.2 | 0.220% |
| Caselaw Access Project | 77.5 | 1 | 77.5 | 19.4 | 1.941% |
| CC Common Crawl | 58.1 | 6 | 348.6 | 87.1 | 8.716% |
| Data Provenance Initiative | 3.4 | 6 | 20.4 | 5.1 | 0.510% |
| Database of Open Access Books | 12 | 6 | 72 | 18 | 1.801% |
| Foodista | 0.08 | 6 | 0.48 | 0.12 | 0.012% |
| GitHub Archive | 40.4 | 6 | 242.4 | 60.6 | 6.064% |
| Library of Congress | 35.6 | 0.25 | 8.9 | 2.2 | 0.220% |
| LibreTexts | 3.6 | 6 | 21.6 | 5.4 | 0.540% |
| News | 0.25 | 6 | 1.5 | 0.38 | 0.038% |
| OERCommons | 0.05 | 6 | 0.3 | 0.08 | 0.008% |

*Continued on next page*

| Source | Size (GB) | Repeats | Effective Size (GB) | Tokens (Billions) | Percentage |
|---|---|---|---|---|---|
| peS2o | 182.6 | 6 | 1,095.6 | 273.9 | 27.409% |
| Pre-1929 Books | 46.3 | 1 | 46.3 | 11.6 | 1.161% |
| PressBooks | 0.6 | 6 | 3.6 | 0.9 | 0.090% |
| Project Gutenberg | 20.1 | 1 | 20.1 | 5 | 0.500% |
| Public Domain Review | 0.007 | 6 | 0.04 | 0.01 | 0.001% |
| PubMed | 147.1 | 1 | 147.1 | 36.8 | 3.683% |
| PEPs | 0.01 | 6 | 0.06 | 0.02 | 0.002% |
| Regulations.gov | 5.1 | 6 | 30.6 | 7.6 | 0.761% |
| StackExchange | 89.7 | 6 | 538.2 | 134.6 | 13.469% |
| Stack V2 | 259.9 | 2 | 519.8 | 130 | 13.009% |
| Ubuntu IRC | 5.3 | 6 | 31.8 | 7.9 | 0.791% |
| UK Hansard | 9.6 | 6 | 57.6 | 14.4 | 1.441% |
| USGPO | 36.1 | 0.25 | 9 | 2.3 | 0.230% |
| USPTO | 661.1 | 0.25 | 165.3 | 41.3 | 4.133% |
| Wikimedia | 57.4 | 6 | 344.4 | 86.1 | 8.616% |
| Wikiteam | 13.7 | 4 | 54.8 | 13.7 | 1.371% |
| CC YouTube | 18.6 | 1 | 18.6 | 4.7 | 0.470% |
| **Total** | **1838.3** | **–** | **3997.4** | **999.3** | **100%** |

## L  Details on Comma's cool-down data mixture

Following Hu et al. [74], we end training with a "cool-down" where we train on 37.7B tokens of high-quality data while linearly decaying the learning rate to 0. We provide the source mixture weights for this cool-down phase in Table 8.

Table 8: Overview of the data mixing used to up/down-weight individual sources in the Common Pile to construct the training distribution for Comma v0.1-1T's cool-down phase. Comma v0.1-2T simply repeats this full mixture twice.

| Source | Size (GB) | Repeats | Effective Size (GB) | Tokens (Billions) | Percentage |
|---|---|---|---|---|---|
| ArXiv Papers | 19.5 | 0.5 | 9.8 | 2.4 | 6.50% |
| CC Common Crawl | 58.1 | 0.3 | 17.4 | 4.4 | 11.63% |

*Continued on next page*

| Source | Size (GB) | Repeats | Effective Size (GB) | Tokens (Billions) | Percentage |
|---|---|---|---|---|---|
| Data Provenance Initiative | 3.4 | 2 | 6.8 | 1.7 | 4.55% |
| Database of Open Access Books | 12 | 2 | 24 | 6 | 16.04% |
| Foodista | 0.08 | 2 | 0.16 | 0.04 | 0.11% |
| LibreTexts | 3.6 | 2 | 7.2 | 1.8 | 0.48% |
| News | 0.25 | 2 | 0.5 | 0.13 | 0.33% |
| OERCommons | 0.05 | 2 | 0.1 | 0.03 | 0.07% |
| peS2o | 182.6 | 0.1 | 18.3 | 4.6 | 12.18% |
| PressBooks | 0.6 | 2 | 1.2 | 0.3 | 0.77% |
| Public Domain Review | 0.007 | 2 | 0.014 | 0.004 | 0.01% |
| PEPs | 0.01 | 2 | 0.02 | 0.005 | 0.02% |
| StackExchange | 89.7 | 0.25 | 22.4 | 5.6 | 14.96% |
| Stack V2 | 259.9 | 0.1 | 26.0 | 6.5 | 17.04% |
| Wikimedia | 57.4 | 0.4 | 23 | 5.7 | 15.32% |
| **Total** | **679.4** | **–** | **149.9** | **37.5** | **100%** |

## M  Details on small-scale data ablations

In subsection 4.3 we report results from a series of small-scale data ablations where we identically trained 1.7B parameter models on various openly licensed and unlicensed datasets and evaluate their performance on the "early signal" tasks from Penedo et al. [133] to compare their data quality against the Common Pile. In Figure 7 we show how the performance of these models evolve over the course of their training run, highlighting that differences in data quality become apparent very early in training. Additionally, we provide exact numerical results for each model in Table 9, showing that the Common Pile has higher data quality than any previously released openly licensed datasets and the Pile, and nearly matches the data quality of the OSCAR dataset. To validate that this is not purely due to the presence of high-quality supervised fine-tuning data from the Data Provenance Initiative (DPI) data source, we also perform an ablation on the Common Pile excluding the DPI data and find that the final performance of this model is largely unchanged.

## N  Additional Comma results

We provide exact numerical results for Comma v0.1-1T and -2T alongside baseline models results across a variety of knowledge, reasoning, and coding tasks in Table 10 and Table 11 respectively. We find that particularly on knowledge-based benchmarks (such as MMLU) and coding benchmarks, Comma v0.1-1T and -2T outperform baseline models trained on an equivalent amount (1T or 2T tokens, respectively) of unlicensed text.

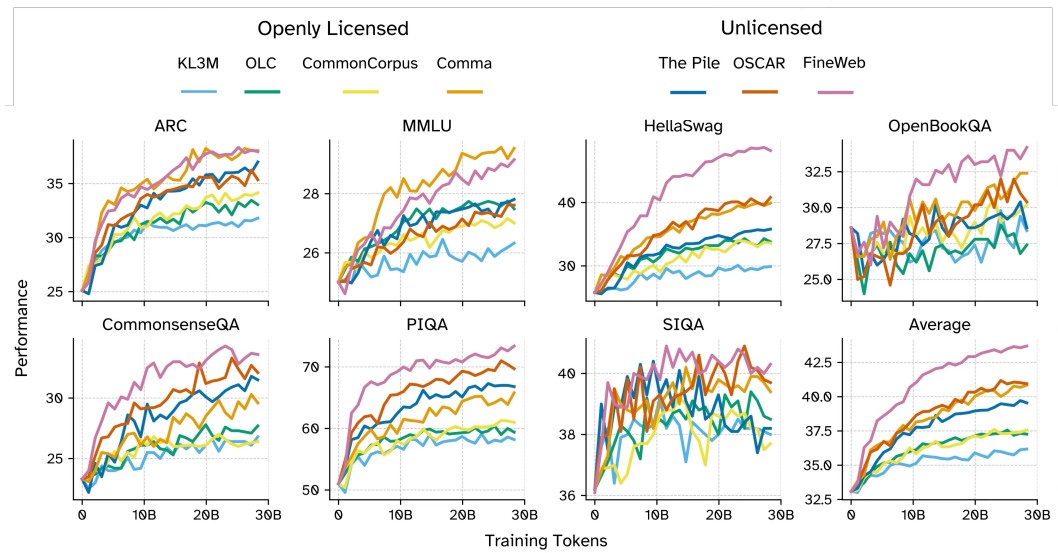

Figure 7: **A model trained on the Comma dataset consistently outperforms models trained on other corpora of openly licensed text and outperforms the Pile on all but two tasks.** We train identical 1.7B parameter models on 28B tokens from each dataset following Penedo et al. [133].

Table 9: **Comma's training dataset has higher quality than previous openly-licensed datasets and unlicensed datasets like the Pile.** In the small-scale (1.7B parameter) data ablation setting, we find that Comma's training dataset yields better models than previous openly licensed datasets and the Pile, and nearly matches the performance of models trained on OSCAR. Additionally, we find that removing the high-quality supervised data from the Data Provenance Initiative has marginal affect on the Comma dataset's overall quality.

| Dataset | ARC | MMLU | HS | OBQA | CSQA | PIQA | SIQA | Avg. |
|---|---|---|---|---|---|---|---|---|
| KL3M | 31.8 | 26.3 | 29.9 | 28.4 | 26.8 | 58.2 | 38.0 | 36.2 |
| OLC | 33.1 | 27.5 | 33.8 | 27.4 | 27.7 | 59.4 | 38.5 | 37.3 |
| Common Corpus | 34.2 | 27.0 | 33.6 | 30.2 | 26.4 | 61.0 | 37.7 | 37.6 |
| Comma (no DPI) | 37.7 | 28.7 | 37.6 | 31.0 | 30.8 | 63.8 | 39.8 | 40.0 |
| Comma | 38.0 | 29.5 | 39.9 | 32.4 | 29.6 | 65.8 | 39.4 | 40.8 |
| The Pile | 37.0 | 27.8 | 35.8 | 28.6 | 31.5 | 66.8 | 38.2 | 39.6 |
| OSCAR | 35.4 | 27.6 | 40.8 | 30.4 | 32.1 | 69.7 | 39.7 | 40.9 |
| FineWeb | 38.0 | 29.1 | 48.2 | 34.2 | 33.6 | 73.4 | 40.3 | 43.7 |

Table 10: Comparison between Comma v0.1-1T and baseline models trained with similar resources (7 billion parameters, 1 trillion tokens) across a variety of knowledge, reasoning, and coding benchmarks. 95% confidence intervals shown.

| Model | ARC-C | ARC-E | MMLU | BoolQ | HS | OBQA | CSQA | PIQA | SIQA | HEval | MBPP | Avg. |
|---|---|---|---|---|---|---|---|---|---|---|---|---|
| RPJ-INCITE | 44.7±2.8 | 66.8±1.9 | 26.4±0.7 | 70.9±1.5 | 70.3±0.9 | 50.2±4.4 | 57.7±2.8 | 77.0±1.9 | 46.4±2.2 | 10.3±4.3 | 17.2±2.9 | 48.9 |
| LLaMA | 48.2±2.9 | 68.2±1.9 | 32.2±0.8 | 75.0±1.5 | 76.2±0.8 | 53.6±4.4 | 61.8±2.7 | 79.3±1.9 | 49.0±2.2 | 17.1±5.2 | 28.1±3.6 | 53.5 |
| StableLM | 43.7±2.8 | 65.8±1.9 | 40.0±0.8 | 70.3±1.6 | 74.3±0.9 | 52.0±4.4 | 57.2±2.8 | 79.0±1.9 | 48.4±2.2 | 24.8±6.1 | 36.5±3.8 | 53.8 |
| MPT | 45.8±2.9 | 67.1±1.9 | 29.4±0.7 | 73.7±1.5 | 76.3±0.8 | 54.0±4.4 | 64.2±2.7 | 80.5±1.8 | 48.9±2.2 | 31.1±6.4 | 35.4±3.8 | 55.1 |
| Open LLaMA | 46.8±2.9 | 67.3±1.9 | 33.1±0.8 | 72.3±1.5 | 74.5±0.8 | 50.8±4.4 | 62.9±2.7 | 79.8±1.8 | 49.5±2.2 | 26.5±6.0 | 33.2±3.7 | 54.2 |
| Comma v0.1-1T | 50.1±2.9 | 68.4±1.9 | 36.0±0.8 | 74.6±1.5 | 64.3±0.9 | 49.8±4.4 | 59.8±2.7 | 72.7±2.0 | 49.3±2.2 | 35.6±6.5 | 34.4±3.8 | 54.1 |
| Qwen3 | 59.7±2.8 | 80.6±1.6 | 73.0±0.7 | 86.6±1.2 | 74.9±0.8 | 52.0±4.4 | 68.1±2.6 | 77.8±1.9 | 51.1±2.2 | 94.3±3.4 | 61.4±4.1 | 70.9 |

Table 11: Performance of Comma v0.1-2T and a variety of budget-matched baseline models, with 95% confidence intervals.

| Model | ARC-C | ARC-E | MMLU | BoolQ | HS | OBQA | CSQA | PIQA | SIQA | HEval | MBPP | Avg. |
|---|---|---|---|---|---|---|---|---|---|---|---|---|
| OLMo Twin | 46.3±2.9 | 65.3±1.9 | 26.0±0.7 | 69.8±1.6 | 74.1±0.9 | 53.4±4.4 | 61.8±2.7 | 79.3±1.8 | 48.5±2.2 | 19.9±5.7 | 23.9±3.4 | 51.7 |
| LLaMA 2 | 50.0±2.9 | 69.3±1.9 | 41.8±0.8 | 77.7±1.4 | 76.0±0.8 | 57.2±4.3 | 62.7±2.7 | 78.8±1.9 | 49.5±2.2 | 26.4±6.1 | 31.1±3.6 | 56.4 |
| Comma v0.1-2T | 51.0±2.9 | 70.6±1.8 | 46.1±0.8 | 79.0±1.4 | 67.8±0.9 | 56.0±4.4 | 64.2±2.7 | 73.0±2.0 | 51.0±2.2 | 44.9±6.7 | 40.7±3.9 | 58.6 |
| DeepSeekLLM | 49.6±2.9 | 68.8±1.9 | 44.2±0.8 | 72.4±1.5 | 76.2±0.8 | 58.0±4.3 | 66.8±2.6 | 79.7±1.8 | 50.9±2.2 | 46.9±6.8 | 43.2±3.9 | 59.7 |

## O   Additional training runs

To explore the sensitivity of our Comma v0.1 results to hyperparameter choices, we perform a series of additional 7B parameter/1T token training runs on AMD MI300A GPUs with slight alterations to the training recipe. Due to both a desire to reach the same 1T token target rapidly, and the lower single-GPU throughput on the system available for these ablations, for all additional runs the the training batch size is 8.3M ($2^{23}$) versus the 2.1M ($2^{21}$) tokens per step of Comma v0.1. Unless otherwise specified, we did not use the two phase training process described in subsection 4.4 (i.e. no separate high-quality cooldown phase is run and we do not perform checkpoint averaging at the end of training and before evaluation).

### O.1   Ablations at 1T Tokens

We first performed a set of training runs for 125,000 steps, resulting in 1.048T total tokens (referred to as "1T" for brevity).

**"8M Batch"**   We perform a run with nearly the same training hyperparameters as Comma v0.1-1T, except with a larger 8M token batch size. We also use a single phase training setup; the base data mixture (Table 7) is run for the entire duration to 1T tokens. The learning rate schedule is 2,000 steps of warm-up from 0 to a peak of $1e-3$ with 123,000 steps of decay to a minimum of $1.8e-9$.

**"Curriculum"**   In this experimental run, a different data mixture is used in each of three training stages of equal duration (we also use the modified hyperparameters from "8M Batch" ablation above). The first stage of the curriculum comprises data from only the Common Pile's largest sources (mostly USPTO, Table 13). The second stage uses the same data mixture as Comma v0.1's main pre-training phase ("phase I"), but run for only 1/3 of the duration. Finally, the third and last stage of the curriculum up-weights Common Pile's highest quality, benchmark-relevant sources (Table 14).

We provide exact numerical results for Comma v0.1 and alternate Comma runs performed with different hyperparameters and data mixture curricula across a variety of knowledge, reasoning, and coding benchmarks in Table 12. We find that the 8M Batch and Curriculum ablations are roughly comparable on average to the main Comma v0.1-1T run, with the notable exception that both ablations slightly outperform Comma v0.1-1T on the coding benchmarks. We conclude that the benchmark results reported for Comma v0.1-1T in subsection 4.4 seem relatively robust to minor changes in training hyperparameters, dataset mixture curriculum (assuming similar amounts of most data splits appear at some time during training), and the software environment and GPU hardware used to train the model.

Table 12: Comparison between our main Comma v0.1-1T training run and alternate runs performed with different hyperparameters and data mixture curricula across a variety of knowledge, reasoning, and coding benchmarks. For "Main", we report the performance of Comma v0.1-1T without averaging the cooldown checkpoints so that it is a fair comparison.

| Model | ARC-C | ARC-E | MMLU | BoolQ | HS | OBQA | CSQA | PIQA | SIQA | HEval | MBPP | Avg. |
|---|---|---|---|---|---|---|---|---|---|---|---|---|
| Curriculum | 45.2 | 69.1 | 41.4 | 74.7 | 60.8 | 46.8 | 59.1 | 70.5 | 48.6 | 38.1 | 34.6 | 53.5 |
| 8M Batch | 47.2 | 69.6 | 42.9 | 69.9 | 62.9 | 47.0 | 56.9 | 70.4 | 50.5 | 36.8 | 37.2 | 53.8 |
| Main | 50.8 | 68.4 | 40.2 | 72.9 | 62.3 | 46.2 | 59.5 | 71.0 | 51.2 | 32.1 | 34.6 | 53.6 |

Table 13: Overview of the data mixing used to up/down-weight individual sources for the Stage 1 of the Curriculum ablation run. In this table we omit the size columns for brevity.

| Source | Repeats | Tokens (Billions) | Percentage |
|---|---|---|---|
| USPTO | 1.4125 | 233.5 | 66.81% |

*Continued on next page*

| Source | Repeats | Tokens (Billions) | Percentage |
|---|---|---|---|
| Pre-1929 Books | 5.65 | 65.4 | 18.71% |
| Stack V2 (HTML) | 11.3 | 12.8 | 3.65% |
| USGPO | 1.41 | 12.8 | 3.65% |
| Library of Congress | 1.41 | 12.6 | 3.59% |
| Biodiversity Heritage Library | 1.41 | 12.52 | 3.58% |
| **Total** | **–** | **349.4** | **100%** |

Table 14: Overview of the data mixing used to up/down-weight individual sources for the Stage 3 of the Curriculum ablation run. In this table we omit the size columns for brevity.

| Source | Repeats | Tokens (Billions) | Percentage |
|---|---|---|---|
| Stack V2 | 1 | 63.8 | 18.519% |
| Database of Open Access Books | 6 | 18 | 5.230% |
| Wikimedia | 6 | 86.1 | 24.981% |
| StackExchange | 2.5 | 56.1 | 16.259% |
| peS2o | 1 | 45.6 | 13.241% |
| CC Common Crawl | 3 | 43.6 | 12.638% |
| ArXiv Papers | 5 | 24.4 | 7.063% |
| Data Provenance Initiative | 6 | 5.1 | 1.485% |
| PressBooks | 6 | 0.87 | 0.251% |
| LibreTexts | 6 | 0.54 | 0.157% |
| News | 6 | 0.37 | 0.108% |
| Foodista | 6 | 0.12 | 0.036% |
| OERCommons | 6 | 0.08 | 0.023% |
| PEPs | 6 | 0.02 | 0.005% |

| Source | Repeats | Tokens (Billions) | Percentage |
|---|---|---|---|
| Public Domain Review | 6 | 0.01 | 0.003% |
| **Total** | **–** | **344.7** | **100%** |

