| Source | Size (GB) | Repeats | Effective Size (GB) | Tokens (Billions) | Percentage |
|---|---|---|---|---|---|
| PressBooks | 0.6 | 2 | 1.2 | 0.3 | 0.77% |
| Public Domain Review | 0.007 | 2 | 0.014 | 0.004 | 0.01% |
| PEPs | 0.01 | 2 | 0.02 | 0.005 | 0.02% |
| StackExchange | 89.7 | 0.25 | 22.4 | 5.6 | 14.96% |
| Stack V2 | 259.9 | 0.1 | 26.0 | 6.5 | 17.04% |
| Wikimedia | 57.4 | 0.4 | 23 | 5.7 | 15.32% |
| **Total** | **679.4** | **–** | **149.9** | **37.5** | **100%** |

## L   Additional Comma Results

We provide exact numerical results for Comma and baseline models across a variety of knowledge and reasoning tasks (Table 10) and coding benchmarks (Table 11). We find that particularly on knowledge-based benchmarks, such as ARC-C and MMLU, and coding benchmarks, Comma outperforms baseline models trained on an equivalent amount (1T tokens) of unlicensed text.

Table 10: Comparison between Comma and baseline models trained with similar resources (7 billion parameters, 1 trillion tokens) across a variety of knowledge and reasoning benchmarks.

| Model | ARC-C | ARC-E | MMLU | BoolQ | HS | OBQA | CSQA | PIQA | SIQA | Avg. |
|---|---|---|---|---|---|---|---|---|---|---|
| RPJ-INCITE | 42.8 | 68.4 | 27.8 | 68.6 | 70.3 | 49.4 | 57.7 | 76.0 | 46.9 | 56.4 |
| Comma | 52.8 | 68.4 | 42.4 | 75.7 | 62.6 | 47.0 | 59.4 | 70.8 | 50.8 | 58.9 |
| MPT | 46.5 | 70.5 | 30.2 | 74.2 | 77.6 | 48.6 | 63.3 | 77.3 | 49.1 | 59.7 |
| OpenLLaMA | 44.5 | 67.2 | 40.3 | 72.6 | 72.6 | 50.8 | 62.8 | 78.0 | 49.7 | 59.8 |
| LLaMA | 44.5 | 67.9 | 34.8 | 75.4 | 76.2 | 51.2 | 61.8 | 77.2 | 50.3 | 59.9 |
| StableLM | 50.8 | 65.4 | 45.2 | 71.7 | 75.6 | 48.2 | 57.2 | 77.0 | 48.2 | 59.9 |
| Qwen3 | 57.2 | 74.5 | 77.0 | 86.1 | 77.0 | 50.8 | 66.4 | 78.2 | 55.0 | 69.1 |

Table 11: Comparison between Comma and baseline models trained with similar resources (7 billion parameters, 1 trillion tokens) on two coding benchmarks.

| Model | HumanEval | MBPP | Avg. |
|---|---|---|---|
| RPJ-INCITE | 11.1 | 15.9 | 13.5 |
| LLaMA | 19.9 | 27.9 | 23.9 |
| StableLM | 23.1 | 32.0 | 27.6 |
| MPT | 27.3 | 33.2 | 30.3 |
| Open LLaMA | 27.6 | 33.9 | 30.8 |
| Comma | 36.5 | 35.5 | 36.0 |
| Qwen3 | 94.5 | 67.5 | 81 |