# OpenReview forum: "The Common Pile v0.1: An 8TB Dataset of Public Domain and Openly Licensed Text"
_NeurIPS.cc/2025/Datasets_and_Benchmarks_Track — NeurIPS 2025 Datasets and Benchmarks Track poster_

### Official Review · Reviewer_Nzh6 · 2025-07-01

**Rating:** 4
**Confidence:** 4

**Summary:**

This paper introduces Common Pile v0.1, an 8TB openly licensed dataset for LLM pre-training, covering 30 diverse sources. To demonstrate its utility, the authors train Comma v0.1, a 7B-parameter model on 1T tokens from the dataset. Despite using only open data, Comma achieves performance competitive with models like LLaMA 7B, especially on knowledge and code benchmarks.

The authors contribute:
1. A publicly released, large-scale, open-license dataset for LLM pretraining;
2. A high-performance model trained on this dataset, demonstrating the viability of ethical and transparent model development;
3. A reproducible pipeline including source selection, data filtering, mixing heuristics, tokenizer training, and model checkpoints.

**Additional Feedback:**

None

**Dataset Code Accessibility:**

Yes

**Dataset Code Comments:**

The authors have made the dataset publicly  available. The dataset is well organized.

**Ethical Considerations:**

No, there are no or only very minor ethics concerns

**Limitations Weaknesses:**

In Section 4.2, the authors emphasize the importance of quality-based data mixing in Comma’s training. However, baseline datasets are evaluated without comparable mixing or rebalancing. This raises questions about whether performance gains stem from the dataset itself or the mixing strategy. The authors should clarify this and, ideally, it would be valuable to include an ablation experiment where Comma is trained without any data mixing.

**Strengths Contributions:**

This paper presents a significant contribution by releasing Common Pile v0.1, an 8TB openly licensed dataset for LLM pretraining, addressing pressing ethical and legal concerns. The dataset is diverse (30 sources) and well-curated.
The authors validate its utility by training Comma v0.1, which achieves competitive performance with models like LLaMA 7B, especially on MMLU, ARC, and code benchmarks.
Compared to prior datasets (OLC, Common Corpus, and KL3M), Common Pile v0.1 is larger, more diverse, and offers better license transparency.
The paper is well-written and clearly structured; figures and tables are informative and support the claims.

---

> ### Author Rebuttal · Authors · 2025-07-31
>
> Thanks so much for your positive feedback. We've responded to the weakness you highlighted below. If you have additional questions, feel free to let us know.
>
> > In Section 4.2, the authors emphasize the importance of quality-based data mixing in Comma’s training. However, baseline datasets are evaluated without comparable mixing or rebalancing. This raises questions about whether performance gains stem from the dataset itself or the mixing strategy. The authors should clarify this and, ideally, it would be valuable to include an ablation experiment where Comma is trained without any data mixing.
>
> Thanks for highlighting this. We opted not to experiment with different mixing weights for the baseline datasets for the following reasons:
> * First, dataset creators generally perform their own data mixing experiments and distribute their datasets with the mixing rates in place. For example, The Pile [1] provides a set of mixing rates and RedPajama [2] uses mixing rates that follow those set by the Llama 1 paper [3]. We therefore believe it is a safe assumption that the distributed mixing weights are performant for a given baseline dataset.
> * Second, for the Common Pile, there is clearly no relationship between a source's size and its quality. For example, patent text from USPTO makes up over a third of the non-code portion of the dataset, but brief inspection of the data indicates that it comprises highly unnatural language (as patents follow a particular format and use standard and very specific terms of legal art) that is likely quite different from most downstream uses. Intuitively, we therefore expected that source mixing rates would be important to tune. However, we surprisingly found relatively little difference in performance when tuning mixing rates. To your point specifically, in our ablation experiments we found that the performance of the "native" mixing rate of just concatenating all sources without rebalancing was comparable to the mixing rates produced by MixMin [4] as well as the hand-tuned rates we ultimately used. Consequently, we expected that performing additional mixing rate tuning for baseline sources would confer limited benefit beyond the mixing rates provided by the baseline datasets' authors.
>
> To support the second point, please find a table below presenting the results (in our ablation experiment setup) of training on Comma's training dataset mixture, the "native" mixture produced by simpling concatenating all sources, and the mixing proportions determined by MixMin. We will include these results in an updated draft in case any readers have similar questions. We hope this better clarifies our choices with regards to mixing rates. Please let us know if you want any further details or justification!
>
> | Method | CSQA | HS | OBQA | PIQA | SIQA | ARC | MMLU | Average |
> |--------|---------------|-----------|------------|------|-----------|------------|-----|------|
> | Native mixing | 0.305 | 0.386 | 0.324 | 0.639 | 0.399 | 0.392 | 0.287 | 0.402 |
> | Hand-tuned mixing | 0.299 | 0.393 | 0.328 | 0.635 | 0.406 | 0.394 | 0.295 | 0.403 |
> | MixMin | 0.298 | 0.393 | 0.316 | 0.651 | 0.413 | 0.394 | 0.290 | 0.401 |
>
> [1] Gao, Leo, et al. "The Pile: An 800GB dataset of diverse text for language modeling." arXiv preprint arXiv:2101.00027 (2020).
>
> [2] Weber, Maurice, et al. "RedPajama: an open dataset for training large language models." Advances in neural information processing systems 37 (2024).
>
> [3] Touvron, Hugo, et al. "Llama: Open and efficient foundation language models." arXiv preprint arXiv:2302.13971 (2023).
>
> [4] Thudi, Anvith, et al. "MixMin: Finding Data Mixtures via Convex Minimization." arXiv preprint arXiv:2502.10510 (2025).

---

> > ### Author Response · Authors · 2025-08-05
> >
> > Hi Reviewer Nzh6, thanks again for your review. We've posted a comprehensive rebuttal that we believe addresses all of the points you raised. If you have additional questions or things to discuss, please let us know. If we have addressed all of your concerns, we'd appreciate if you considered raising your score accordingly. Thanks again!

---

> > ### Comment · Area_Chair_6deh · 2025-08-06
> > **Urgent Reminder: Reviewer Discussion**
> >
> > Dear Reviewer Nzh6,
> >
> > Thank you for support to NeurIPS! This is a reminder that Reviewer-Author discussions is extended by 48h till Aug 8, 11.59pm AoE.
> >
> > Since you have not participated in the discussion, I want to reiterate the following key points:
> >
> > 1. Reviewers are expected to stay engaged in discussions,
> > - It is not OK to stay quiet.
> > - It is not OK to leave discussions till the last moment.
> > - If authors have resolved your (rebuttal) questions, do tell them so.
> > - If authors have not resolved your (rebuttal) questions, do tell them so too.
> >
> > 2.	Please note “Mandatory Acknowledgement” button is to be submitted only when reviewers fulfill all conditions below (conditions in the acknowledgment form):
> > - Read the author rebuttal
> > - Engage in discussions (reviewers must talk to authors, and optionally to other reviewers and AC - ask questions, listen to answers, and respond to authors)
> > - Fill in "Final Justification" text box and update “Rating” accordingly (this can be done upon convergence - reviewer must communicate with authors first)
> >
> > Finally, please treat authors the way you would like to be treated (fairness, politeness, calmness, attention and focus on merits). Thanks again for your time and effort.
> >
> > Best regards,
> >
> > AC

---

### Official Review · Reviewer_LNc3 · 2025-07-01

**Rating:** 5
**Confidence:** 5

**Summary:**

This paper presents the Common Pile v0.1, an 8TB dataset comprising text from 30 openly licensed sources, designed for training large language models. The authors claim this constitutes the largest assembly of openly licensed text to date. They validate their dataset by training Comma v0.1, a 7B parameter model, demonstrating competitive performance against models trained on unlicensed datasets. The work addresses important ethical and legal concerns in LLM training by focusing exclusively on public domain and openly licensed content.

**Additional Feedback:**

**Technical Soundness**: The experimental methodology follows established protocols (Penedo et al.) and the training procedures are well-documented and reproducible.

**Presentation Quality**: The paper is well-written with clear figures and comprehensive appendices. The licensing discussion in Section 2 is particularly well-crafted and accessible.

**Impact Potential**: This work has significant potential to influence how the field approaches training data curation, potentially establishing new standards for ethical AI development.

**Suggestions for Improvement**:

1. More prominent discussion of scale claims limitations in abstract/introduction
2. Expanded evaluation on larger model scales
3. Discussion of strategies for maintaining licensing compliance over time
4. Additional analysis of domain coverage gaps

**Dataset Code Accessibility:**

Yes

**Ethical Considerations:**

No, there are no or only very minor ethics concerns

**Final Justification:**

I maintain my score at **5**.

The rebuttal effectively clarifies existing points but doesn't elevate the contribution beyond my original assessment. The dataset/document licensing distinction validates their scale claims, and performance gaps are contextualized rather than resolved.

While the clarifications are helpful, they address concerns already factored into my Accept rating. The limitations (performance gaps, no large-scale evaluation) remain. The core contribution—an 8TB openly licensed dataset—was already recognized as significant in my initial review.

No new strengths emerged that would warrant raising to Strong Accept. The work remains technically solid with important ethical implications, justifying Accept but not higher.

**Limitations Weaknesses:**

**Scale Claims Require Careful Examination**: The claim of being "the largest assembly of openly licensed text to date" needs more nuanced framing. While the paper compares against OLC (0.85TB), Common Corpus (7.4TB), and KL3M (3TB), the comparison with RedPajama-V2 requires clarification. RedPajama-V2 contains 30T tokens of web data, but as the authors correctly note, collection licenses don't necessarily align with underlying document licenses. However, this distinction should be made more prominent in the abstract and introduction to avoid potential misunderstanding.

**Limited Diversity in Some Domains**: The authors acknowledge that performance lags on HellaSwag and PIQA, attributing this to limited coverage of personal blogs, tutorials, and hobbyist content. This limitation suggests the dataset may not be optimal for all downstream applications.

**Temporal Licensing Concerns**: The paper acknowledges that licensing information may become outdated as content creators change licenses post-collection. This presents an ongoing challenge for the dataset's long-term viability that could be addressed more thoroughly.

**Evaluation Scope**: While the 7B model comparison is valuable, evaluation on larger scales (e.g., 13B+ parameters) would strengthen claims about the dataset's potential for training state-of-the-art models.

**Strengths Contributions:**

**Significant Contribution to Ethical AI Development**: The paper addresses a critical gap in responsible AI development by creating a large-scale dataset that respects intellectual property rights. This work is timely given the ongoing legal challenges facing LLM developers regarding copyright infringement.

**Comprehensive Licensing Framework**: The authors demonstrate rigorous attention to licensing details, implementing strict due diligence standards and excluding sources with questionable licensing (e.g., OpenAlex, Hacker News on Kaggle). The adherence to the Open Knowledge Foundation's Open Definition 2.1 provides a clear, principled framework.

**Thorough Empirical Validation**: The controlled comparison experiments using the Penedo et al. methodology provide convincing evidence of dataset quality. The demonstration that Comma v0.1 achieves competitive performance with LLaMA 7B, MPT 7B, and RedPajama-INCITE 7B under similar computational budgets is particularly compelling.

**Methodological Rigor**: The paper provides detailed documentation of data collection, filtering, and processing procedures. The discussion of data mixing strategies and the distinction between the raw Common Pile and the processed "Comma dataset" shows careful experimental design.

**Reproducibility and Openness**: The release of the complete dataset, processing code, and model checkpoints exemplifies the open science principles the work advocates for.

---

> ### Author Rebuttal · Authors · 2025-07-31
>
> Thanks for your positive feedback and thorough review. We have answered your questions and addressed your concerns below. Please let us know if any additional information would be helpful.
>
> > The claim of being "the largest assembly of openly licensed text to date" needs more nuanced framing. While the paper compares against OLC (0.85TB), Common Corpus (7.4TB), and KL3M (3TB), the comparison with RedPajama-V2 requires clarification.
>
> Thanks for flagging your uncertainty here. We stand by the claim that the Common Pile is the largest collection of public domain and openly licensed text. We bring up RedPajama-V2 in the paper as an example of a dataset with an ODC BY license. In our experience, people sometimes confuse dataset-level licenses like ODC BY with document-level licenses like CC BY. However, the use of an ODC BY license does not imply that the documents comprising the dataset are themselves openly licensed. Therefore, datasets like RedPajama-V2 should not be considered "openly licensed", and there is no direct comparison to draw between the size of the Common Pile and RedPajama-V2. We will review our wording in the paper to make sure this is made clear.
>
> > The authors acknowledge that performance lags on HellaSwag and PIQA, attributing this to limited coverage of personal blogs, tutorials, and hobbyist content. This limitation suggests the dataset may not be optimal for all downstream applications.
>
> Thank you for highlighting this limitation. While our "dataset ablation" results do indicate significant gaps on some tasks, we want to emphasize that other datasets often specifically tailor filtering to improve performance on these datasets. For example, FineWeb's dataset curation pipeline was specifically designed to maximize performance on the benchmarks on which we measure performance. In particular, the FineWeb paper [1] discusses the steps specifically taken to improve performance on HellaSwag. Moreover, anecdotal experience suggests that because HellaSwag only covers two narrow domains, HellaSwag can be gamed by upweighting those domains (which does not improve performance on other benchmarks). We did not tune our dataset filtering pipeline to maximize performance on specific benchmarks, which could further explain the difference in performance. Furthermore, while we agree that the Common Pile might not be optimal for all downstream applications, we do want to highlight that the Common Pile does appear to work best in certain applications (e.g. knowledge-intensive tasks). We will provide further discussion of this in our updated draft.
>
> > Temporal Licensing Concerns: The paper acknowledges that licensing information may become outdated as content creators change licenses post-collection. This presents an ongoing challenge for the dataset's long-term viability that could be addressed more thoroughly.
>
> For licenses we consider, when one obtains a piece of content under a specific license, they acquire it subject to the terms of that license at the time of acquisition. For example, Creative Commons licenses come with specific conditions that apply perpetually once the license is granted – the Creative Commons (CC) FAQ states that "CC licenses are not revocable… As a licensor, you may stop distributing under the CC license at any time, but anyone who has access to a copy of the material may continue to redistribute it under the CC license terms." We will update our paper to include this language.
>
> > Evaluation Scope: While the 7B model comparison is valuable, evaluation on larger scales (e.g., 13B+ parameters) would strengthen claims about the dataset's potential for training state-of-the-art models.
>
> We agree that training larger scale models would be interesting, but we currently lack the financial and computational resources to do so (Comma v0.1 used about  $100,000 USD of compute). However, we anticipate that the Common Pile will prove equally valuable for training larger scale models and hope to explore this in future work.
>
> [1] Penedo, Guilherme, et al. "The fineweb datasets: Decanting the web for the finest text data at scale." Advances in Neural Information Processing Systems 37 (2024): 30811-30849.

---

> > ### Author Response · Authors · 2025-08-05
> >
> > Hi Reviewer LNc3, thanks again for your review. We've posted a comprehensive rebuttal that we believe addresses all of the points you raised. If you have additional questions or things to discuss, please let us know. If we have addressed all of your concerns, we'd appreciate if you considered raising your score accordingly. Thanks again!

---

> > ### Comment · Area_Chair_6deh · 2025-08-06
> > **Urgent Reminder: Reviewer Discussion**
> >
> > Dear Reviewer LNc3,
> >
> > Thank you for support to NeurIPS! This is a reminder that Reviewer-Author discussions is extended by 48h till Aug 8, 11.59pm AoE.
> >
> > Since you have not participated in the discussion, I want to reiterate the following key points:
> >
> > 1. Reviewers are expected to stay engaged in discussions,
> > - It is not OK to stay quiet.
> > - It is not OK to leave discussions till the last moment.
> > - If authors have resolved your (rebuttal) questions, do tell them so.
> > - If authors have not resolved your (rebuttal) questions, do tell them so too.
> >
> > 2.	Please note “Mandatory Acknowledgement” button is to be submitted only when reviewers fulfill all conditions below (conditions in the acknowledgment form):
> > - Read the author rebuttal
> > - Engage in discussions (reviewers must talk to authors, and optionally to other reviewers and AC - ask questions, listen to answers, and respond to authors)
> > - Fill in "Final Justification" text box and update “Rating” accordingly (this can be done upon convergence - reviewer must communicate with authors first)
> >
> > Finally, please treat authors the way you would like to be treated (fairness, politeness, calmness, attention and focus on merits). Thanks again for your time and effort.
> >
> > Best regards,
> >
> > AC

---

> > ### Comment · Reviewer_LNc3 · 2025-08-08
> >
> > Thank you for the clarifications.
> >
> > Your responses effectively address my concerns—the licensing distinction is well-articulated and the performance gap context is helpful. However, these clarifications reinforce rather than exceed my original assessment.
> >
> > I'm maintaining my Accept score because while your rebuttal resolves ambiguities, it doesn't reveal new strengths beyond what I initially recognized. The limitations (HellaSwag/PIQA gaps, 7B-only evaluation) remain, though adequately explained.

---

### Official Review · Reviewer_Z9hp · 2025-07-06

**Rating:** 5
**Confidence:** 4

**Summary:**

This paper presents the Common Pile v0.1, an 8TB dataset of openly licensed text designed for LLM pre-training, comprising content from 30 diverse sources. The authors validate their dataset by training Comma v0.1, a 7B parameter LLM on 1 trillion tokens, demonstrating competitive performance with models trained on unlicensed data of similar computational budgets. The work addresses growing concerns about copyright infringement and ethical issues in the collection of LLM training data.

**Additional Feedback:**

Suggestions for Improvement:
- Include human evaluation or more diverse automatic metrics to capture potential quality differences better
- Discuss plans for dataset updates as licensing information changes
- More detailed analysis of how performance might scale with additional openly licensed sources

Questions for Authors:
1. How sensitive are the results to the specific mixing weights chosen?
2. Have you considered partnerships with content creators to expand openly licensed content?
3. What are the computational cost trade-offs compared to web-scraped approaches?
4. Given that licensing information can change post-collection, what legal framework governs the continued use of data whose license status has become ambiguous or been revoked?
5. When combining sources with different but compatible open licenses (e.g., CC BY and CC BY-SA), how do you ensure the resulting model respects the most restrictive terms, and could this create attribution obligations for model users?
6. Does this work provide empirical evidence that could inform legal debates about whether LLM training constitutes transformative use under the fair use doctrine, given the demonstrable differences in model behavior between licensed and unlicensed training regimes?

**Dataset Code Accessibility:**

Yes

**Dataset Code Comments:**

The authors demonstrate exceptional commitment to reproducibility:
- Both raw Common Pile v0.1 and processed training mixture are available
- All data collection and processing code is released
- Comma v0.1 checkpoints and training configurations are provided
- Comprehensive documentation enables replication

**Ethical Comments:**

This work represents a positive contribution to AI ethics rather than raising concerns:

Positive Ethical Impact:
- The explicit focus on openly licensed content addresses fundamental IP concerns in AI training
- Comprehensive licensing metadata and source documentation enable proper attribution
- PII redaction and toxicity filtering demonstrate responsible data handling

Appropriate Safeguards:
- Rigorous license verification processes with manual validation of top domains
- Conservative approach to synthetic data and uncertain licensing
- Transparent acknowledgment of remaining uncertainties

Minor Considerations:
- While consent wasn't explicitly obtained from content creators, open licensing represents implicit consent for the use cases described.
- The environmental impact of large-scale training is mentioned in the checklist, but could be discussed more prominently.

**Ethical Considerations:**

No, there are no or only very minor ethics concerns

**Limitations Weaknesses:**

Performance Gaps:
- Comma significantly underperforms on HellaSwag, PIQA, and CommonSenseQA compared to models trained on unlicensed data. The authors acknowledge this may reflect domain coverage limitations, but the gaps are substantial (e.g., ~10 points on HellaSwag vs. FineWeb).
- While impressive for openly licensed data, the 8TB dataset is still smaller than current industry-standard datasets, which limits its long-term competitiveness.

Technical Limitations:
- The paper lacks error bars or confidence intervals due to computational constraints. Single-run results make it difficult to assess the reliability of the results.
- Despite careful curation, the authors acknowledge that license laundering and detection errors may still occur. This fundamental challenge limits the dataset's claims to legal certainty.
- The benchmark suite, while standard, doesn't capture all aspects where licensed vs. unlicensed data might differ (e.g., creativity, nuanced reasoning).

Methodological Concerns:
- The mixing weight determination process (Section 4.2) is somewhat ad-hoc, relying on heuristic performance assessments rather than principled optimization.
- The dataset collection was completed in late 2024, but licensing information may become outdated, creating ongoing maintenance challenges.

**Strengths Contributions:**

- This work tackles one of the most pressing ethical and legal challenges in AI development - the use of copyrighted material for LLM training. The timing is particularly relevant given ongoing litigation and industry concerns.
- At 8TB from 30 diverse sources, this represents the largest openly licensed text dataset to date, substantially larger than previous efforts, such as OLC (0.85TB), and more diverse than KL3M.
- The authors establish clear criteria based on the Open Knowledge Foundation's Open Definition 2.1, with careful attention to license laundering and due diligence (Section 2.1).
The filtering, deduplication, and mixing methodology is well-documented and reproducible, utilizing established tools like Dolma and adhering to best practices.
- The controlled comparison across multiple datasets (Section 4.3), using identical training setups, provides strong evidence for the quality of the datasets.
- Comma v0.1 outperforms budget-matched baselines on multiple benchmarks, particularly excelling on knowledge-based tasks (MMLU, ARC-C) and code tasks.
- The paper is well-written and organized, with clear descriptions of the methodology and comprehensive appendices.
- Figures and tables effectively communicate results and dataset composition.
- The authors provide transparent discussion of limitations and caveats throughout the paper.

---

> ### Author Rebuttal · Authors · 2025-07-31
>
> Thank you for your encouraging feedback and points for discussion. We have answered your questions and addressed your concerns below. Please let us know if you'd like further clarification; we are happy to have additional discussion about our work.
>
> > gaps are substantial (e.g., ~10 points on HellaSwag vs. FineWeb).
>
> Thank you for highlighting this limitation. While our "dataset ablation" results indicate significant gaps on some tasks, we want to emphasize that FineWeb's dataset curation pipeline was specifically designed to maximize performance on the benchmarks on which we measure performance. In particular, the FineWeb paper [1] discusses the steps specifically taken to improve performance on HellaSwag. Moreover, anecdotal experience suggests that because HellaSwag only covers two narrow domains, HellaSwag can be gamed by upweighting those domains (which does not improve performance on other benchmarks). We did not tune our dataset filtering pipeline to maximize performance on specific benchmarks, which could further explain the difference in performance. We will further highlight this in our updated draft.
>
> > the 8TB dataset is still smaller than current industry-standard datasets, which limits its long-term competitiveness.
>
> While we agree that the Common Pile v0.1 is likely too small to support extremely long training durations like those used in the recent Qwen3 models, we would argue that it can still support large-scale training runs. To support this assertion, between submission time and now we trained a new variant of Comma (Comma v0.1-2T) trained on two trillion tokens from the Common Pile by simply duplicating the training data for the original 1 trillion token training run. We can then directly compare to budget-matched 7 billion parameter models trained on 2 trillion tokens. We find, for example, that Comma v0.1-2T outperforms the budget-matched Llama 2 7B [2] on average (full results in our [response to reviewer MAt9](https://openreview.net/forum?id=DIELgiqdvJ&noteId=UlmXJBLkZk) -- omitted here due to character limits). We have added the Comma v0.1-2T results to our draft.
>
> Additionally, as indicated by our v0.1 designation, we believe there is a large amount of untapped openly licensed and public domain data that could be included in future versions, such as the recently released Institutional Books 1.0 corpus [4] of one million scanned public-domain books. To further support the possibility of larger scales, we analyzed the sources in the Common Pile that contain publication year metadata and found that the quantity of public domain and openly licensed is increasing superlinearly, with roughly half of our corpus released since 2020. We will add this content to an appendix and with a reference and brief discussion in the main text.
>
> > The paper lacks error bars or confidence intervals
>
> We agree that confidence intervals could strengthen the robustness of our claims. While performing multiple training runs at the scales we consider would be prohibitively expensive, we sought to address this concern by re-running our evaluation using the lm-evaluation-harness library [5], which includes functionality for computing bootstrapped confidence intervals by systematically omitting portions of each evaluation dataset. We found that the reported confidence intervals were generally relatively small compared to the performance differences, supporting the significance of performance differences between various models. For example, on MMLU where Comma v0.1 outperforms the next-best budget-matched model by over 6%, the confidence interval is only 0.4%. We will include these updated results in our paper..
>
> > license laundering and detection errors may still occur. This fundamental challenge limits the dataset's claims to legal certainty.
>
> Thank you for highlighting that our dataset cannot be claimed to be free of license laundering and other errors - this is an important point that we also highlighted in Section 2.1. We do want to emphasize that we are not promising legal certainty in our work. We will review our paper to ensure this is made crystal clear and edit accordingly. Apart from possible licensing errors, there are many intersecting legal issues currently being litigated and reasoned about by lawyers and regulators concerning the training of LLMs. In light of unsettled law and differences between jurisdictions globally, it would be irresponsible to promise legal certainty. Even if the law were settled on these matters, our work is an academic paper whose primary contribution is in machine learning research. As a research artifact, we do not intend for our work to be providing legal advice.
>
> > The benchmark suite ... doesn't capture all aspects
>
> While we agree that our benchmark suite might not capture all aspects of a base model's performance, we note that the collection of tasks is widely accepted as comprehensive and sufficient for comparing base models, and has been used in many past works (e.g. [1, 2, 6] and many others). If there are other benchmarks that you suggest we include, please let us know and we can add them.
>
> > The mixing weight determination process is somewhat ad-hoc
>
> Thanks for bringing up our methodology for choosing mixing weights. Please note that we experimented extensively with MixMin [7], a principled approach to optimizing mixing weights. Our final hand-tuned mixing weights were informed by those generated by MixMin, but were simplified for easier reproducibility. As mentioned in our paper, performance was not very sensitive to the exact mixing ratios, and there was limited benefit in using the exact mixing rates provided by MixMin (40.1% vs 40.3% on average). We have added a new appendix section that includes the full results for different mixing strategies.
>
> > licensing information may become outdated
>
> For licenses we consider, when one obtains a piece of content under a specific license, they acquire it subject to the terms of that license at the time of acquisition. For example, Creative Commons licenses come with specific conditions that apply perpetually once the license is granted – the Creative Commons (CC) FAQ states that "CC licenses are not revocable… As a licensor, you may stop distributing under the CC license at any time, but anyone who has access to a copy of the material may continue to redistribute it under the CC license terms." We will update our paper to include this language.
>
> > The environmental impact ... could be discussed more prominently.
>
> Thanks for encouraging us to highlight the approximate environmental impact of the Comma v0.1 training run. Throughout the 18-day-long training of Comma v0.1, all 64 H100 GPUs retained high utilization and therefore the maximal 700W of power usage. Comma v0.1 was trained in the us-east1 zone of Amazon Web Services for 18 days, where approximately 593.4 pounds of CO2 are emitted per megawatt-hour. Consequently, we estimate the training of Comma v0.1 to have consumed about 11,000 pounds (5,200 kg) of CO2eq. We will add these estimates to our paper.
>
> > Have you considered partnerships with content creators to expand openly licensed content?
>
> It would be incredible to partner with creators to help release their content with an open license! We think creators would be most persuaded if we could provide convincing evidence that their content would be valuable for language model training. We hope the results in our paper satisfy this need and plan to reach out to creators in the future.
>
> > What are the computational cost trade-offs compared to web-scraped approaches?
>
> For sources that we must crawl and scrape ourselves (e.g., our News source) the costs are similar to web scraping, but for sources where text is already available (e.g., Project Gutenberg) there are minimal to no computational costs. Overall the computational costs of collecting our sources are minuscule compared to training data filtering or LM training, so we did not report them, but if you think it would be helpful we can add approximate costs.
>
> > how do you ensure the resulting model respects the most restrictive terms, and could this create attribution obligations for model users?
>
> Whether a model itself (as distinct from its output) is a derivative work of its training data is a legal question that has not been definitively answered by U.S. courts. Consequently, there is no established precedent as to whether and how a model or its users must respect the training data's license (via attribution obligations or otherwise). We will update our draft to emphasize that we do not mean to convey any conclusion about the derivative works question or any other unanswered question of law.
>
> > Does this work provide empirical evidence that could inform legal debates?
>
> While we hope this work enables more analysis at the intersection of copyright and machine learning, we are not legal scholars and this analysis is beyond the scope of our work. However, we do want to note that our paper indicates that there may *not* be demonstrable differences in model behavior when training on openly licensed or unlicensed data.
>
> [1] Penedo, Guilherme, et al. "The fineweb datasets: Decanting the web for the finest text data at scale." NeurIPS 37 (2024): 30811-30849.
>
> [2] Touvron, Hugo, et al. "Llama 2: Open foundation and fine-tuned chat models." arXiv:2307.09288 (2023).
>
> [3] Hoffmann, Jordan, et al. "Training compute-optimal large language models." arXiv:2203.15556 (2022).
>
> [4] Cargnelutti, Matteo, et al. "Institutional Books 1.0: A 242B token dataset from Harvard Library's collections, refined for accuracy and usability." arXiv:2506.08300 (2025).
>
> [5] Gao, Leo, et al. "The Language Model Evaluation Harness". Zenodo 10.5281/zenodo.12608602 (2024).
>
> [6] Groeneveld, Dirk, et al. "Olmo: Accelerating the science of language models." arXiv:2402.00838 (2024).
>
> [7] Thudi, Anvith, et al. "MixMin: Finding Data Mixtures via Convex Minimization." arXiv:2502.10510 (2025).

---

> > ### Author Response · Authors · 2025-08-05
> >
> > Hi Reviewer 79hp, thanks again for your review. We've posted a comprehensive rebuttal that we believe addresses all of the points you raised. If you have additional questions or things to discuss, please let us know. If we have addressed all of your concerns, we'd appreciate if you considered raising your score accordingly. Thanks again!

---

> > ### Comment · Area_Chair_6deh · 2025-08-06
> > **Urgent Reminder: Reviewer Discussion**
> >
> > Dear Z9hp,
> >
> > Thank you for support to NeurIPS! This is a reminder that Reviewer-Author discussions is extended by 48h till Aug 8, 11.59pm AoE.
> >
> > Since you have not participated in the discussion, I want to reiterate the following key points:
> >
> > 1. Reviewers are expected to stay engaged in discussions,
> > - It is not OK to stay quiet.
> > - It is not OK to leave discussions till the last moment.
> > - If authors have resolved your (rebuttal) questions, do tell them so.
> > - If authors have not resolved your (rebuttal) questions, do tell them so too.
> >
> > 2.	Please note “Mandatory Acknowledgement” button is to be submitted only when reviewers fulfill all conditions below (conditions in the acknowledgment form):
> > - Read the author rebuttal
> > - Engage in discussions (reviewers must talk to authors, and optionally to other reviewers and AC - ask questions, listen to answers, and respond to authors)
> > - Fill in "Final Justification" text box and update “Rating” accordingly (this can be done upon convergence - reviewer must communicate with authors first)
> >
> > Finally, please treat authors the way you would like to be treated (fairness, politeness, calmness, attention and focus on merits). Thanks again for your time and effort.
> >
> > Best regards,
> >
> > AC

---

### Official Review · Reviewer_MAt9 · 2025-07-07

**Rating:** 5
**Confidence:** 4

**Summary:**

This paper introduces The Common Pile v0.1, one 8TB openly licensed english text corpus designed to serve as a large-scale, open, and diverse dataset for language model pretraining. The dataset aggregates content from a wide range of sources, including web crawls, academic papers, books, and other public datasets, with a focus on maximizing coverage, diversity, and data quality. The paper details the data collection, deduplication, filtering, and documentation processes, and provides statistics and analyses to characterize the corpus. The authors position The Common Pile as a resource for the research community, aiming to facilitate reproducible and transparent model development. Authors also use The Common Pile as the single data source to train a  language model - Comma v0.1, which achieve competitive performance compared to LLMs trained on top of unlicensed dataset (e.g. LLaMA-1 7B, MPT 7B, 58
and RedPajama-INCITE 7B).

**Additional Feedback:**

This is good work. However, this is only in small scale and I hope in the near future, we could have more large-scale datasets like this.

**Dataset Code Accessibility:**

Yes

**Dataset Code Comments:**

The code, data and model checkpoints are all open-sourced.

**Ethical Considerations:**

No, there are no or only very minor ethics concerns

**Limitations Weaknesses:**

**W1:** For the data mixing experiment, could you report the performance with difference data mix ratio as well as the result from MixMin?
When comparing with other datasets as a pretraining dataset, how the other dataset's data mixing ratio is tuned?
Also, here is some sota work fro determining data mix ratio:
- regmix: data mixture as regression for language model pre-training

**W2:** I didn't find the corresponding implementation for data preprocessing including different filtering and deduplication techniques in the github repo?

**W3:** Could you justify why Comma v0.1 is underperforming LLMs trained with other licensed datasets on OBQA, CSQA and PIQA?

**W4:** The current scale of Common Pile v0.1 can only support small LLM training.

**Strengths Contributions:**

**S1:** This work is a valuable and timely contribution to the NLP/LLM community in terms of preserving creator
rights and legal transparency, offering a well-documented, large-scale openly licensed text dataset covering a wide range of domain (Scientific and scholarly texts, Source Code, Wikis, etc)

**S2:**  The authors provide a thorough comparison to LLMs pretrained on top of existing datasets such as KL3M, OLC, and OSCAR, highlighting both overlaps and unique aspects. The distinction from prior work is clearly articulated and justified.

**S3:** The dataset preprocessing part is comprehensive with different techniques applied, including text quality classifier, likelihood-260
based filtering and regex filtering.

**S4:** The training receipt for Comma v0.1 is well-documented, which provides enough details for reproduction.

---

> ### Author Rebuttal · Authors · 2025-07-31
>
> Thank you for your encouraging feedback and helpful suggestions. We've responded to your questions and concerns below. Please let us know whether our responses and updates address your concerns or if you would like additional clarification and information.
>
> > W1: For the data mixing experiment, could you report the performance with difference data mix ratio as well as the result from MixMin?
>
> Thanks for highlighting that these additional experimental results could be helpful to include. We have added a new appendix section to our updated draft that includes the results copied below. As mentioned in our paper, we found that performance was not very sensitive to the exact mixing ratios, and that there was limited benefit in using the exact mixing rates provided by MixMin.
>
> | Method | CSQA | HS | OBQA | PIQA | SiQA | ARC | MMLU | Average |
> |--------|---------------|-----------|------------|------|-----------|------------|-----|------|
> | MixMin | 0.298 | 0.393 | 0.316 | 0.651 | 0.413 | 0.394 | 0.290 | 0.401 |
> | Comma v0.1 mixing | 0.299 | 0.393 | 0.328 | 0.635 | 0.406 | 0.394 | 0.295 | 0.403 |
>
> > When comparing with other datasets as a pretraining dataset, how the other dataset's data mixing ratio is tuned?
>
> Since we only compare to other datasets that have been proposed as LLM pre-training datasets, we use the native mixing ratios provided by the creators of each dataset. For example, The Pile [1] provides a set of mixing rates and RedPajama [2] uses mixing rates that follow those set by the Llama 1 paper [3]. We therefore believe it is a safe assumption that the distributed mixing weights are performant for a given baseline dataset. While it's possible that we could improve performance on other datasets slightly by tuning mixing ratios, we note again that our results on tuning mixing rates on the Common Pile indicate that performance is relatively insensitive to changes in mixing strategies. Consequently, we are confident that our results show that the Common Pile compares favorably to a wide range of reasonable off-the-shelf baselines that have been previously used to train performant LLMs. We will clarify this in our updated draft.
>
> > Also, here is some sota work fro determining data mix ratio: regmix: data mixture as regression for language model pre-training
>
> Thank you for bringing up RegMix. We decided not to explore using RegMix because the MixMin paper [4] compares extensively to RegMix and demonstrates improved performance while being simpler and computationally cheaper. Additionally, the limited gains (compared to heuristically tuned mixing rates) of MixMin suggests that we might not expect additional improvement from using different data mixing algorithms, so we did not continue to explore along these lines. We will mention RegMix and this decision in our updated draft.
>
> > W2: I didn't find the corresponding implementation for data preprocessing including different filtering and deduplication techniques in the github repo?
>
> Thanks for pointing out our oversight in not linking to our data processing code. Our publicly available code repository includes the Dolma Toolkit tagging, deduplication, and mixing configuration files along usage instructions on how to reproduce the filtered versions of each source in the Common Pile v0.1. We have been told not to post non-anonymized links in our response, but we will include a link to our data processing code in our updated draft.
>
> > W3: Could you justify why Comma v0.1 is underperforming LLMs trained with other licensed datasets on OBQA, CSQA and PIQA?
>
> Since our experimental results focus on evaluating base models (i.e., without post-training) trained on incredibly large text corpora, it is difficult to pinpoint why different pre-training corpora result in different performance on certain downstream benchmarks (apart from very simple insights like "comma performs better on coding benchmarks due to the inclusion of a significant amount of high-quality code"). However, we do note that OBQA, CSQA, and PIQA all require some degree of commonsense knowledge, so it's possible that our corpus covers less of this kind of information. We might conjecture that the heavier reliance on "specialized" (e.g. research papers, patents, etc.) text limits exposure to more general commonsense knowledge, though this is only a conjecture. We will further highlight and discuss this discrepancy in our updated draft.
>
> > W4: The current scale of Common Pile v0.1 can only support small LLM training.
>
> While we agree that the Common Pile v0.1 is likely too small to support extremely long training durations like those used in the recent Qwen3 models, we would argue that it can still support large-scale training runs. To support this assertion, between submission time and now we trained a new variant of Comma (called Comma v0.1-2T) trained on two trillion tokens from the Common Pile by simply duplicating the training data for the original 1 trillion token training run. We can then directly compare to budget-matched seven billion parameter models trained on two trillion tokens. The results are shown in the table below. Notably, even at this larger scale, Comma v0.1-2T remains competitive with models trained on unlicensed data, such as Llama 2 7B [5]. We note that two trillion tokens is about 15x longer than Chinchilla-optimal at this parameter scale [6]. While we currently lack the computational resources to train models with larger parameter counts, we suspect that the Common Pile could support training larger models. We have added the Comma v0.1-2T results to our updated draft.
>
> | Model | ARC-C | ARC-E | MMLU | BoolQ | HS | OBQA | CSQA | PIQA | SIQA | HEval | MBPP | Avg. |
> |-------|-------|-------|------|-------|----|----- |------|------|------|-------|------|------|
> | OLMo Twin        | 45.2 | 67.5 | 28.2 | 71.7 | 73.4 | 48.0 | 61.8 | 77.9 | 48.5 | 18.2 | 27.5 | 51.6 |
> | Llama 2              | 48.5 | 69.5 | 45.8 | 80.2 | 76.2 | 48.4 | 62.8 | 76.7 | 50.8 | 26.1 | 28.5 | 55.8 |
> | Comma v0.1 2T | 45.8 | 71.8 | 49.8 | 78.6 | 64.4 | 46.2 | 64.0 | 72.5 | 52.3 | 44.2 | 41.5 | 57.4 |
> | DeepSeekLLM   | 49.5 | 67.7 | 48.5 | 71.7 | 74.1 | 52.0 | 66.6 | 77.8 | 51.6 | 43.1 | 43.8 | 58.8 |
>
> > I hope in the near future, we could have more large-scale datasets like this.
>
> We completely agree that future work should be done to expand our efforts - this is why we chose to designate our work with the v0.1 version! We believe there is a large amount of untapped openly licensed and public domain data that could be included in future versions, such as the recently released Institutional Books 1.0 corpus [7] of one million scanned public-domain books. To further support the possibility of larger scales, we analyzed the sources in the Common Pile that contain publication year metadata and found that the quantity of public domain and openly licensed is increasing superlinearly, with roughly half of our corpus released since 2020. We will add this content to an appendix in our updated draft and with a reference and brief discussion in the main text.
>
> [1] Gao, Leo, et al. "The Pile: An 800GB dataset of diverse text for language modeling." arXiv preprint arXiv:2101.00027 (2020).
>
> [2] Weber, Maurice, et al. "RedPajama: an open dataset for training large language models." Advances in neural information processing systems 37 (2024).
>
> [3] Touvron, Hugo, et al. "Llama: Open and efficient foundation language models." arXiv preprint arXiv:2302.13971 (2023).
>
> [4] Thudi, Anvith, et al. "MixMin: Finding Data Mixtures via Convex Minimization." arXiv preprint arXiv:2502.10510 (2025).
>
> [5] Touvron, Hugo, et al. "Llama 2: Open foundation and fine-tuned chat models." arXiv preprint arXiv:2307.09288 (2023).
>
> [6] Hoffmann, Jordan, et al. "Training compute-optimal large language models." arXiv preprint arXiv:2203.15556 (2022).
>
> [7] Cargnelutti, Matteo, et al. "Institutional Books 1.0: A 242B token dataset from Harvard Library's collections, refined for accuracy and usability." arXiv preprint arXiv:2506.08300 (2025).

---

> > ### Author Response · Authors · 2025-08-05
> >
> > Hi Reviewer MAt9, thanks again for your review. We've posted a comprehensive rebuttal that we believe addresses all of the points you raised. If you have additional questions or things to discuss, please let us know. If we have addressed all of your concerns, we'd appreciate if you considered raising your score accordingly. Thanks again!

---

> > ### Comment · Area_Chair_6deh · 2025-08-06
> > **Urgent Reminder: Reviewer Discussion**
> >
> > Dear MAt9,
> >
> > Thank you for support to NeurIPS! This is a reminder that Reviewer-Author discussions is extended by 48h till Aug 8, 11.59pm AoE.
> >
> > Since you have not participated in the discussion, I want to reiterate the following key points:
> >
> > 1. Reviewers are expected to stay engaged in discussions,
> > - It is not OK to stay quiet.
> > - It is not OK to leave discussions till the last moment.
> > - If authors have resolved your (rebuttal) questions, do tell them so.
> > - If authors have not resolved your (rebuttal) questions, do tell them so too.
> >
> > 2.	Please note “Mandatory Acknowledgement” button is to be submitted only when reviewers fulfill all conditions below (conditions in the acknowledgment form):
> > - Read the author rebuttal
> > - Engage in discussions (reviewers must talk to authors, and optionally to other reviewers and AC - ask questions, listen to answers, and respond to authors)
> > - Fill in "Final Justification" text box and update “Rating” accordingly (this can be done upon convergence - reviewer must communicate with authors first)
> >
> > Finally, please treat authors the way you would like to be treated (fairness, politeness, calmness, attention and focus on merits). Thanks again for your time and effort.
> >
> > Best regards,
> >
> > AC

---

### Comment · Area_Chair_6deh · 2025-08-03
**Reminder for Reviewer-Author discussions**

Dear reviewers,

Thank you so much for all your time and effort supporting NeurIPS!

If you haven't yet, please take a moment to read through the author's rebuttal. The reviewer-author discussion period is crucial for ensuring a fair and comprehensive evaluation of their work. If the rebuttal addresses your concerns, please acknowledge this and adjust your scores accordingly. If not, please let them know which concerns remain and if you have any follow-up questions. Your thoughtful feedback will help authors improve their scholarship and propel our field forward.

I know this is a busy time, and really appreciate your effort.

Best Regards
Area Chair

---

### Note · Authors · 2025-08-12

We thank the reviewers again for their questions and comments. While we only heard from one of the reviewers during the rebuttal period, we firmly believe we have addressed every concern and question raised. We provide a brief summary of our responses to some common points below.

**Comparing mixing rates**

Multiple reviewers asked for the performance of our "ablation" models using the mixing weights provided by MixMin and/or using the "native" mixing weights resulting from simply concatenating all of the sources. While we did not include these results in our original draft, we found that there was limited difference in performance from tuning the mixing weights, and the MixMin, native, and "hand-tuned" weights all performed comparably. We designed and chose the hand-tuned weights because they are easy to implement, which facilitates reproducibility.

**Scaling up**

Some reviewers wondered whether the Common Pile v0.1 would support larger scale training runs. To answer this, we trained Comma v0.1-2T simply by training on the 1T Comma v0.1 training dataset twice. We found it performed comparably to other budget-matched unlicensed models and even matched the performance of Qwen 3 8B (trained for 36T tokens) on many benchmarks. We unfortunately lack the computational resources to train larger models (in terms of parameter count), but these results suggest there is no reason that the Common Pile could not support training runs of larger models. Additionally, we performed an analysis of the publication dates of content in the Common Pile and found that the amount of openly licensed and public domain content has been steadily growing and that about half of the documents in the Common Pile were published in the last 5 years, supporting the growth of the Common Pile in future iterations.

**Per-benchmark gaps**

We also were asked for more details about the lower performance of Common Pile-based models on datasets like HellaSwag in the ablation setup. Beyond the simple fact that the Common Pile does not cover the same set of domains as web text-derived datasets, in our rebuttal we additionally clarified that datasets like FineWeb have filtering pipelines that are specifically designed to improve performance on datasets like HellaSwag (see Penedo et al. 2024, section 3.5). Consequently, performance gaps compared to those datasets should be expected.

Updates have been made to our draft reflect the above points. Thanks again for your time and consideration.

---

### Decision · Program_Chairs · 2025-09-18

**Decision:**

Accept (poster)

**Comment:**

This paper releases the Common Pile v0.1, an 8TB collection of openly licensed text designed for LLM pre-training. It comprises content from 30 sources that span diverse domains including research papers, code, books, encyclopedias, educational materials, audio transcripts, and more. This paper also release the code used in dataset creation as well as Comma v0.1’s checkpoints and training mixture. Based on this dataset, authors train Comma v0.1 a 7 billion parameter LLM trained on 1 trillion tokens. Comma attains competitive performance to LLMs trained on unlicensed text with similar computational budgets, such as LLaMA 7B. The paper details the data collection, deduplication, filtering, and documentation processes, and provides statistics and analyses to characterize the corpus.

Strengths:

1.	This work is a valuable and timely contribution to the NLP/LLM community in terms of preserving creator rights and legal transparency, offering a well-documented, large-scale openly licensed text dataset covering a wide range of domain.
2.	The authors provide a thorough comparison to LLMs pretrained on top of existing datasets such as KL3M, OLC, and OSCAR, highlighting both overlaps and unique aspects.
3.	The paper provides detailed documentation of data collection, filtering, and processing procedures.

Weaknesses:

1.	Comma v0.1 is underperforming LLMs trained with other licensed datasets on OBQA, CSQA and PIQA.
2.	Comma v0.1is a is smaller than current industry-standard datasets. It is likely too small to support extremely long training durations like those used in the recent Qwen3 models.
3.	The licensing information may be outdated as content creators change licenses post-collection. This presents an ongoing challenge for the dataset's long-term viability

Reasons:

Prior data collection efforts have yielded datasets too small or low-quality to produce performant LLMs. This openly licensed dataset can provide support for future research in this filed.

Discussions:

There are relatively few discussions for this paper. Reviewers have general agreement regarding both the advantages and disadvantages, which are shown above.